



# Localized injections of interactive volcanic aerosols and their climate impacts in a simple general circulation model

Joseph P. Hollowed[1], Christiane Jablonowski[1], Hunter Y. Brown[2], Benjamin R. Hillman[2], Diana L. Bull[2], and Joseph L. Hart[2]

[1]Department of Climate and Space Sciences and Engineering, University of Michigan, Ann Arbor, MI, USA
[2]Sandia National Laboratories, Albuquerque, NM, USA

**Correspondence:** Joseph Hollowed (hollowed@umich.edu)

**Abstract.** A new set of standalone parameterizations is presented for simulating the injection, evolution, and radiative forcing by stratospheric volcanic aerosols against an idealized Held-Suarez-Williamson atmospheric background in the Energy Exascale Earth System Model version 2. Sulfur dioxide ($SO_2$) and ash are injected into the atmosphere with a specified profile in the vertical, and proceed to follow a simple exponential decay. The $SO_2$ decay is modeled as a perfect conversion to a long-living sulfate aerosol which persists in the stratosphere. All three species are implemented as tracers in the model framework, and transported by the dynamical core's advection algorithm. The aerosols contribute simultaneously to a local heating of the stratosphere and cooling of the surface by a simple plane-parallel Beer-Lambert law applied on two zonally-symmetric radiation broadbands in the longwave and shortwave range. It is shown that the implementation parameters can be tuned to produce realistic temperature anomaly signatures of large volcanic events. In particular, results are shown for an ensemble of runs that mimic the volcanic eruption of Mt. Pinatubo in 1991. The design requires no coupling to microphysical subgrid-scale parameterizations, and thus approaches the computational affordability of prescribed-aerosol forcing strategies. The idealized simulations contain a single isolated volcanic event against a statistically uniform climate, where no background aerosols or other sources of externally-forced variability are present. This model configuration represents a simpler-to-understand tool for the development of climate source-to-impact attribution methods.

## 1 Introduction

Volcanic eruptions are one of the most dominant natural sources of exogenous forcing on the Earth system. In large volcanic events, the stratosphere can be loaded with extraordinary amounts of sulfur dioxide ($SO_2$), which gradually oxidize to form long-living sulfate aerosols (Bekki, 1995). In the case of tropical eruptions, the radiative properties of long-living aerosols subsequently lead to global stratospheric and surface-level temperature deviations up to a few degrees Kelvin from climatological averages, which can persist for years (Kremser et al., 2016; McCormick et al., 1995; Dutton and Christy, 1992). Variations in the stratospheric sulfate content by the Earth's volcanic history has thus been one of the strongest drivers of interannual climate variability (e.g. Schurer et al. (2013)).



Since volcanic eruptions impact the climate, there is a rich history of implementing volcanic forcing parameterizations for coupled Earth system models (ESMs) in the literature. Simpler techniques prescribe radiative aerosol properties directly from an external dataset or analytic forms (e.g., see Toohey et al. (2016); Eyring et al. (2013); Gao et al. (2008); Kovilakam et al. (2020)). Prescribed forcing approaches might be chosen for their computational affordability, though they are also used to facilitate climate model intercomparisons by standardizing the forcing scheme (Zanchettin et al., 2016). More complex approaches prescribe emissions of volcanic $SO_2$, which are then handed to separate aerosol, chemistry, and advection codes. These codes then explicitly model the aerosol evolution, transport, and radiative properties (e.g., Mills et al. (2016, 2017); Brown et al. (2024)). A review of these modeling choices for volcanic forcings is presented in Marshall et al. (2022).

Prescribed and prognostic methods have also been applied to model other forms of sulfur-based radiative forcing, with significant research recently being devoted to stratospheric aerosol injection (SAI) climate-change intervention activities (Crutzen, 2006; Tilmes et al., 2018, 2017; McCusker et al., 2012). In addition, there is growing interest in solving the "attribution problem" of quantifying causal connections between an observed climate impact, and an upstream forcing source. Volcanoes are a natural analog to SAI, and thus offer a pathway for developing novel attribution methods.

The climate impacts that are most relevant to society, such as droughts, heat waves, or fires, are multiple steps away from their associated sources (e.g. volcanoes, or other solar radiation modification). Therefore, there is a need for robust multi-step attribution techniques in both climate change studies (Burger et al., 2020) and climate intervention studies (National Academies of Sciences, 2021; Office of Science and Technology Policy (OSTP), 2023). Multi-step attribution involves a sequence of data analyses that connect a source to a downstream impact with specific assessments of each step (Hegerl et al., 2010). Examples of multi-step attribution are uncommon, with the storyline approach from the extreme weather attribution community coming the closest (Trenberth et al., 2015; Shepherd, 2016; Pettett and Zarzycki, 2023).

As the climate community increasingly relies on advanced statistical inference and machine learning approaches to attribute downstream impacts, it is critical to develop testbeds which can be widely shared and used to understand the accuracy of the methods' inferences. Although the development of verification datasets for advanced data analytic techniques in the climate community is nascent, there are a few examples. Fulton and Hegerl (2021) generated synthetic climate modes to test the accuracy of distinct pattern extraction techniques and show that the most commonly used principal component analysis technique does not perform well. Mamalakis et al. (2022) worked to develop an "attribution benchmark dataset" for which the ground truth is known to enable evaluation of different explainable artificial intelligence (AI) methods.

Currently, developing data analytic methods for multi-step attribution in the context of volcanic forcing is restricted to models that utilize expensive prognostic aerosol treatments. This is because with prescribed forcing approaches in free-running atmospheric simulations, there is a dynamical inconsistency between the transport patterns and aerosol distributions. In particular, the forcing dataset does not respond to the atmospheric state. Accordingly, we suggest that a useful testbed for the attribution problem between stratospheric aerosol forcing and atmospheric temperature perturbations could be built upon a new idealized representation of a large volcanic eruption event within a highly simplified atmospheric environment.

Here we outline a simulation strategy which enables an affordable prognostic aerosol implementation for idealized climate model configurations. Our design seeks to maintain a realistic spatio-temporal signature of the atmospheric impacts, while





minimizing the terms contributing to temperature and wind tendencies as much as possible. The former is achieved by including a localized injection and subsequent transport of aerosols by a tracer advection scheme. The latter is achieved by coupling the
aerosol concentrations directly to the temperature field. While traditional approaches often require the inclusion of an auxiliary radiative transfer code for this second step, our implementation is standalone.

Our approach sacrifices realism by design. The goal is not to accurately replicate any particular historical eruption, or to asses a model based on its specific post-eruption climate predictions, but rather to produce a plausible realization of a volcanic event, simulated with a minimal forcing set. Although our configuration does not offer a deterministic answer to the
attribution problem, it facilitates foundational attribution research by representing key process characteristics between a source and downstream impact, and can provide large datasets without the typical computational burden of climate simulations. This work thereby supports the development and testing of novel data analyses and attribution techniques.

Our model isolates a single volcanic event from any other external source of forcing or variability, and allows the flexibility to be embedded in a simplified atmospheric environment. Specifically, we describe the implementation of an eruption similar
in character to the 1991 eruption of Mt. Pinatubo, and the subsequently observed impacts (Karpechko et al., 2010; Robock, 2000; McCormick et al., 1995; Hansen et al., 1992) in an idealized so-called Held-Suarez-Williamson (HSW; Williamson et al. (1998)) configuration of the Energy Exascale Earth System Model version 2 (E3SMv2; Golaz et al. (2022)). The HSW configuration on a flat earth replaces E3SMv2's physical parameterization package with a temperature relaxation towards a prescribed, hemispherically-symmetric equilibrium temperature and Rayleigh friction near the surface. These two forcing
mechanisms mimic radiative effects and the boundary-layer turbulence, respectively. There are no background aerosols, no moisture, and no long-term climate trends. The implementation of the injection, aerosol dissipation, and forcing can be tuned to yield sensible atmospheric impacts for almost any model configuration with qualitatively realistic circulation patterns, even in absence of a standard physical parameterization suite.

The paper is structured as follows. The simplified climate model configuration of E3SMv2 is described in Sect. 2. Section 3
introduces the idealized volcanic injection, sulfate formation, and radiative forcing parameterizations. This is followed by a discussion of the ensemble design, simulation results, and the computational expense in Sect. 4. Section 5 summarizes the findings and provides an outlook on their utility for the modeling community. Appendix A describes custom modifications that were needed for our chosen simplified climate implementation with E3SMv2. In addition, Appendix B provides recommendations for the tuning of the suggested aerosol parameterizations.

## 85  2  Climate Model Configuration

When choosing the base model configuration, the goal was to provide an environment in which the volcanic forcing can be nearly isolated. In addition, we aimed at keeping the number of physical subgrid-scale forcing mechanisms small. These simplifications are achieved by running a climate model in an atmosphere-only mode, and replacing the standard suite of physical parameterizations with simple forcing functions for the temperature and horizontal winds.





Section 2.1 introduces the E3SMv2 climate model which serves as the foundation for our research. E3SMv2's chosen HSW configuration is a modified implementation of the idealized forcing set originally described by Held and Suarez (1994) (hereafter HS94), involving a damping of low-level winds and a relaxation of the temperature field to a specified zonally-symmetric reference profile, described in Sect. 2.2. The main difference between the HS94 and HSW forcing is the presence of a more realistic relaxation temperature profile above 100 hPa which generates stratospheric polar jets in the HSW variant. Section 2.3 describes a simple extension for idealized physics packages which provides global, zonally-symmetric longwave and shortwave radiation profiles.

### 2.1 The E3SMv2 climate model

E3SMv2 is a state-of-the-art climate model that consists of various coupled components for the atmosphere, ocean, land, sea ice, and land ice (Golaz et al., 2022). The dynamical core of the E3SM Atmosphere Model version 2 (EAMv2) uses a spectral-element (SE) solver on a quasi-uniform cubed-sphere grid for a shallow, hydrostatic atmosphere (Taylor et al., 2020), and a semi-Lagrangian tracer transport scheme (Bradley et al., 2022) which ensures local mass conservation and shape preservation. Specifically, the experiments presented here use the $\sim 2°$ "ne16pg2" grid, where each cubed-sphere element features a 2x2 grid of physics columns. The grid for the physical parameterizations is thus coarser than the associated dynamics grid (Hannah et al., 2021; Herrington et al., 2019). The vertical grid consists of 72 vertical levels with a model top at 0.1 hPa, or approximately 60 km.

We use a highly simplified, dry configuration of EAMv2 with no topography, no moisture, and no coupling to other components. The physical parameterization suite is replaced by the idealized forcing set. Internally, this configuration is labeled as the "FIDEAL" component set— an inheritance of E3SMv2 from its original fork of the Community Earth System Model (CESM, Danabasoglu et al. (2020)). As a part of our work, the FIDEAL component set needed to be revived, and is not functional in the official release of E3SMv2.

We note that the ne16pg2 grid is coarser than the default E3SMv2 $\sim 1°$ "ne30pg2" grid. We have not tested activating our implementation on such a higher-resolution grid, which will require a re-tuning the model parameters. Recommendations for performing the tuning of the volcanic forcing are given in Appendix B.

### 2.2 Idealized climate forcing

The HS94 forcing was originally proposed as a benchmark for the intercomparison of statistically steady-states produced by the dry dynamical cores of Atmospheric General Circulation Models (AGCMs) without topography. The forcing includes the Rayleigh damping of low-level winds to represent friction in the boundary layer and a Newtonian temperature relaxation toward an analytic "radiative equilibrium" temperature $T_{\text{eq}}(\phi, p)$ given by

$$\frac{\partial T}{\partial t} = \ldots - k_T(\phi, p) \left[ T - T_{\text{eq}}(\phi, p) \right]. \tag{1}$$

Here, $T$ is the temperature, $t$ stands for the time, $\phi$ represents the latitude, $p$ symbolizes the pressure, and $k_T(\phi, p)$ is the relaxation rate. $T_{\text{eq}}$ has no time dependence and therefore does not include any diurnal or seasonal cycles. The temperature variability



on any timescale is purely driven by the internal dynamics that is nudged towards the equilibrium. The form of the equilibrium temperature is designed to mimic the net effects of radiation, convection, and other subgrid-scale processes. Williamson et al. (1998) (hereafter W98) later noted that since the HS94 benchmark deliberately maintains a "passive" stratosphere, sup-

porting none of the typical stratospheric structures such as the polar jets, it would not be applicable to their dynamical core intercomparison studies of tropopause formation. To remedy this deficiency, they provide a modification of the original HS94 equilibrium temperature, which includes realistic lower-stratospheric lapse rates in the tropics and polar regions. Such a HSW configuration was, e.g., also used in Yao and Jablonowski (2016) who explored Sudden Stratospheric Warmings (SSWs) in the idealized environment.

We use the HSW forcing in our simulations, and omit all other physical parameterizations. This setup provides an atmosphere that is characterized by realistic dynamical motions and a quasi-realistic idealized climatology while maintaining a highly simplified inventory of diabatic subgrid forcings. In implementing HSW in E3SMv2, a few notable modifications were made. First, the lapse rate of the equilibrium temperature $T_{eq}$ was set to zero above 2 hPa to maintain realistic upper-stratospheric temperatures. Next, in addition to the HS94 treatment of surface friction, we include a second Rayleigh damping mechanism

near the model top as a "sponge layer" for calming the polar jet winds and absorbing spurious wave reflections, as described in Jablonowski and Williamson (2011). Specifics of these HSW modifications are provided in Appendix A.

Figure 1 shows the equilibrium temperature in the latitude-pressure plane, the vertical profile of the wind damping strength, and the resulting 10-year average zonal-mean temperature and zonal wind fields following a five-year spinup period using these idealized forcings in E3SMv2. The resulting stratospheric temperature and wind structures are quasi-realistic, reaching

maximum tropical temperatures of about 240 K at the 50-60 km height levels which correspond to the region between 1-0.1 hPa. However, these temperatures are slightly cooler than the observed values near 50 km (1 hPa) which are about 260 K, as documented in Fleming et al. (1990). Temperature minima are seen near the tropical tropopause, as well as the polar middle-stratosphere. Sharp vertical temperature gradients are seen near the polar upper-stratosphere, leading to temperatures in excess of 270 K. In the zonal wind, we see the formation of tropospheric mid-latitude westerly jets with maximum wind speeds

of $\sim$30 m s$^{-1}$, and strong stratospheric polar jets in excess of 60 m s$^{-1}$. Easterlies up to -30 m s$^{-1}$ dominate the tropical stratosphere. As there are no seasonal variations present, each hemisphere eternally varies about this winter-like steady state, which is qualitatively representative of observations (Fleming et al., 1990), while the symmetric thermally-direct circulation (Hadley cells) is more consistent with equinox states in nature (see discussion in Sect. 4.2).

## 2.3  Extending the HSW model with simple radiation

The HSW forcing set is unaware of radiative processes, except by the extent to which they are mimicked in the temperature relaxation toward $T_{eq}$. Energy balance at the top of the atmosphere (TOA) is implied, though there are no specifications of incoming or outgoing radiative fluxes.

However, in computing the diabatic heating and cooling terms of stratospheric aerosols in Sect. 3, it will be both convenient and natural to have expressions for the flux densities of incoming shortwave (SW) and outgoing longwave (LW) broadbands,

which are qualitatively consistent with the HSW equilibrium temperature field. We first define a global, zonally-symmetric



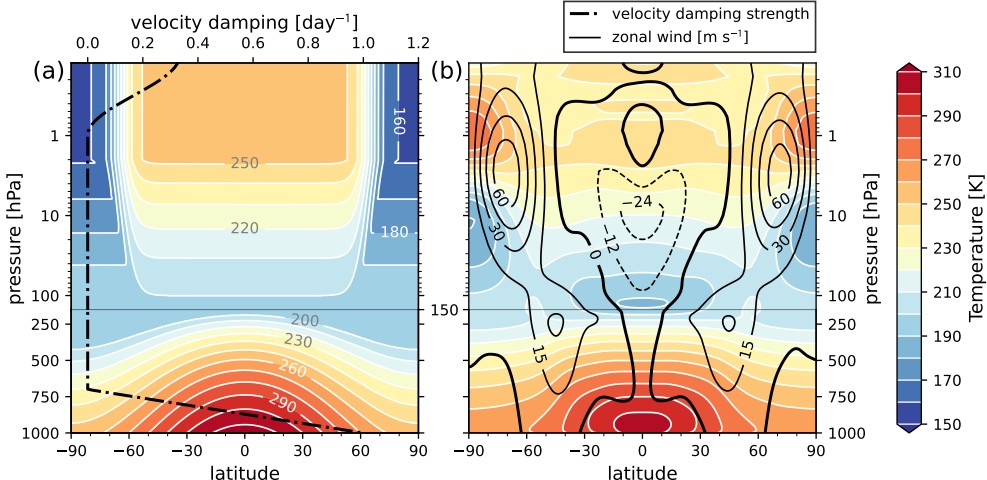

**Figure 1. (a)** The modified HSW equilibrium temperature in the latitude-pressure plane. Contours are drawn every 10 K. Overlaid as a thick dashed black line is the vertical profile of the velocity damping coefficient for both the sponge layer and the surface, with its values on the top horizontal axis (see Appendix A for details). **(b)** 10-year average zonal-mean temperature and zonal wind distributions in an E3SMv2 run with temperature relaxation toward the reference temperature of panel (a), after a five-year spinup period. Temperature contours are drawn every 10 K, while positive (negative) wind contours every 15 m s$^{-1}$ (12 m s$^{-1}$). Negative contours are dashed, and the zero-line is shown in bold. For all variables shown, the vertical (pressure) axis is logarithmic above 150 hPa, and linear below 150 hPa. The separation between these two domains given as gray horizontal lines.

longwave flux density based on $T_{\text{eq}}$ at the surface, and then deduce a shortwave component by setting the total integrated global power equal to that of the longwave component. Both flux density profiles will be constant in time.

In both the HS94 and HSW models, the radiative equilibrium temperature below 100 hPa is

$$T_{\text{eq}}(\phi, p) = \max\left[ (200\text{ K}), \left[ (315\text{ K}) - (60\text{ K})\sin^2\phi - (10\text{ K})\log\left(\frac{p}{p_0}\right)\cos^2\phi \right] \left(\frac{p}{p_0}\right)^{R_d/c_p} \right], \tag{2}$$

where $R_d/c_p = 2/7$ is the ratio of the ideal gas constant and specific heat at constant pressure for dry air. At the reference pressure $p_0 = 1000$ hPa, the equation reduces to

$$T_{\text{eq}}(\phi, p_0) = 315\text{ K} - (60\text{ K})\sin^2\phi. \tag{3}$$

We compute a longwave graybody flux density $I_{\text{LW}}$ from the Stefan-Boltzman law as

$$I_{\text{LW}} = \sigma T_{\text{surf}}^4 = \sigma\left[ 315\text{ K} - (60\text{ K})\sin^2\phi \right]^4 \tag{4}$$

where $\sigma$ is the Stefan-Boltzman constant. If desired, $T_{\text{surf}}$ can be the actual surface temperature on the 2D surface mesh. We instead choose a simplified approach that is both analytic and static in time, by approximating the surface temperature



as Eq. (3). For incident shortwave radiation, we use a simple cosine form which vanishes at the poles, resembling equinox conditions, given by

$$I_{\text{SW}} = I_0 \cos \phi. \tag{5}$$

By integrating Eq. (4) and Eq. (5) over the sphere, we find that a normalization parameter of $I_0 \approx 560$ W m$^{-2}$ enforces that the total globally-integrated power is in balance between $I_{\text{LW}}$ and $I_{\text{SW}}$. We note that these radiative fluxes are considerably higher than the annual average solar insolation of the real Earth system. The primary reason for the enhanced values is that there is no attenuation of the upwelling longwave radiation by moisture, clouds, or other background constituents (excluding volcanic aerosols) in the HSW atmosphere. Including such an effect in the HSW configuration would be arbitrary and overly-complicated. Further, we will show in Sect. 3 that the aerosol radiative forcing design has sufficient freedom in the number of tunable parameters to achieve desired heating rates, without being preferential about the amplitudes of $I_{\text{SW}}$ and $I_{\text{LW}}$.

The resulting flux profiles are shown in Fig. 2. This figure shows an energy deficit poleward of $55°$, and a surplus equatorward, with maxima in the net flux in the midlatitudes. We emphasize that the shape of the flux profiles is the important aspect here, and that balancing $I_{\text{LW}}$ and $I_{\text{SW}}$ is only being done for style and physical legibility. This "radiation" will be used *only* to control the heating and cooling rates imposed by injected aerosols, which will ultimately be subject to model tuning, and will have no effect on mean atmospheric temperatures. The overall climate and energy balance is still controlled independently by the HSW temperature relaxation.

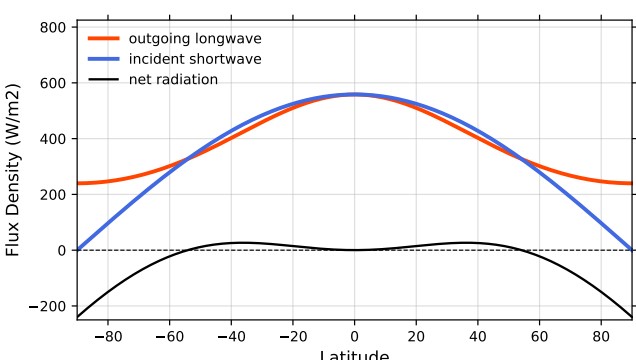

**Figure 2.** Longwave (Eq. (4)), shortwave (Eq. (5)), and net flux densities as functions of latitude.

## 3 The idealized volcanic forcing model

We model radiative forcing by stratospheric aerosol injection events in the idealized HSW environment by directly forcing the temperature field via a standalone parameterization. This forcing is done without the need for intermediary aerosol or radiation models. While this approach could be generalized for any local injections of sulfur species to the atmosphere, the



implementation used here is designed and tuned to produce a realistic representation of the 1991 eruption of Mt. Pinatubo. Specifically, our model describes the localized simultaneous injection of volcanic ash and sulfur dioxide ($SO_2$) with a specified vertical profile over a single model column. Decay of the $SO_2$ in turn leads to production of long-lived sulfate aerosols. These chemical species are implemented as "tracers" within EAMv2 (scalar mixing ratio quantities advected by the model's transport scheme), and contribute independently to local and surface temperature tendencies.

The strategy is to add together various ingredients as follows: (1) define volcanic sources (stratospheric injection), (2) define $SO_2$ sinks (sulfate aerosol production), (3) compute the aerosol optical depth (AOD) of each model column, and (4) increment the temperature tendency (local radiative heating by absorption, and radiative surface cooling by AOD). Steps (1)-(4) are described in Sect. 3.1-3.4, respectively. Section 3.5 provides a brief summary of the model, a table of the model parameters, and notes on the parameter tuning strategy.

### 3.1 Tracer injection

We model the time tendency of each injected tracer species $j$ ($SO_2$ and ash) as the sum of a source and sink:

$$\frac{\partial m_j}{\partial t} = R(m_j) + f. \tag{6}$$

$R(m_j)$ is an exponential removal function with e-folding timescale $1/k_j$, and the source term $f$ describes the spatial distribution of injection:

$$R(m_j) = -k_j m_j, \tag{7}$$

$$f = \widetilde{A}_j T(t) H(\phi, \lambda) V(z). \tag{8}$$

The separable temporal, vertical, and horizontal dependencies are $T(t)$, $V(z)$, and $H(\phi, \lambda)$, respectively, where $z$ is the geometric height and $\lambda$ symbolizes the longitude. $\tilde{A}_j$ is a normalization parameter. This form is then discretized onto the model grid with horizontal column, vertical level, and timestep indices $i$, $k$, and $n$, respectively. We choose to model an injection uniformly distributed over a single column and time period of length $\delta t$. Explicitly, the "injection region" $S$ is symbolized as

$$S = \{(\phi_i, \lambda_i, t_n) \mid \phi_i = \phi_{i'}, \ \lambda_i = \lambda_{i'}, \ t_0 \leq t_n \leq t_0 + \delta t\}, \tag{9}$$

where $i'$ is the index of the injection column. The product of $T(t_n)$ and $H(\phi_i, \lambda_i)$ in Eq. (8) is then replaced by an indicator function $I_{i,n} \equiv I(t_n, \phi_i, \lambda_i)$, which is equal to one inside of $S$, and equal to zero outside of $S$. The mass tendency is then

$$\frac{\partial m_{j,i,k}}{\partial t} = -k_j m_{j,i,k} + I_{i,n} A_j V(z_k). \tag{10}$$

The source $f$ for tracer $j$ is normalized by the constant $A_j$, which scales the *total* injected mass to a known parameter $M_j$, by

$$M_j = A_j \delta t \sum_k V(z_k) \tag{11}$$

$$\implies A_j = \frac{M_j}{\delta t \sum_k V_k}, \tag{12}$$



where we define $V_k \equiv V(z_k)$. The normalization constant $A_j$ is unique to the vertical grid configuration, and converges to the normalization of the analytic form $A_j \to \widetilde{A}_j$ with increasing resolution. This treatment avoids losing mass to numerical diffusion once the injected mass is placed onto the model grid.

Rather than a mass tendency, the quantity required by EAMv2's physics interface is the tendency of the tracer mixing ratio $q_j \equiv m_j/m_{\mathrm{atm}}$. If $\Delta p_{i,k}$ is the local pressure thickness of the grid cell, $g$ is acceleration due to gravity, and $a_i$ is the column

area, the air mass $m_{\mathrm{atm}}$ in this definition can be replaced via the hydrostatic equation, yielding

$$q_{j,i,k} = m_{j,i,k} \, \frac{g}{\Delta p_{i,k} \, a_i} \; . \tag{13}$$

Given Eq. (10)-(13), the final expression implemented for the update of tracer $j$ at column $i$, vertical level $k$, and time $n$ is

$$\frac{\partial q_{j,i,k,n}}{\partial t} = \frac{g}{\Delta p_{i,k} \, a_i} \left[ -k_j \, m_{j,i,k} + I_{i,n} \, \frac{M_j}{\delta t \sum_k V_k} V_k \right] \; . \tag{14}$$

For the vertical dependence $V(z)$, we follow Fisher et al. (2019) and assume a Gaussian distribution defined by a center of

mass altitude $\mu$, and a geometrical standard deviation

$$V(z) = \exp \left( -\frac{1}{2} \frac{(z - \mu)^2}{(1.5 \; \mathrm{km})^2} \right) . \tag{15}$$

The profile deviation of 1.5 km is a compromise between Fisher et al. (2019) and the width of the parabolic injection profile of Stenchikov et al. (2021). In hydrostatic models such as E3SMv2, the height $z$ is a diagnostic quantity. Therefore, the vertical profile needs to be computed at each timestep. We inject both SO$_2$ and ash at the same height and with the same deviation, and

we do not include a normalization coefficient in $V(z)$, since $A_j$ is already scaled by $\sum_k V_k$.

We tuned the center of mass altitude $\mu$ by ensuring that the sulfate tracer which eventually arises from the initial SO$_2$ injection settles in the lower stratosphere, between 20 and 25 km, consistent with estimates from aerosol transport models (Sheng et al., 2015) and forcing reconstructions (see review in Sect. 2 of Toohey et al. (2016)). As in previous works, we might expect to inject just above the tropical tropopause, near $\mu = 17\text{-}18$ km (Stenchikov et al., 2021; Fisher et al., 2019), and then

allow the self-lofting process to carry the plume to a level of neutral buoyancy in the lower stratosphere. Per Stenchikov et al. (2021), this process is expected to be driven by a vertical velocity of $w \approx 1$ km day$^{-1}$ due to strong initial radiative heating rates of about 20 K day$^{-1}$ in the dense, fresh volcanic plume. After tuning the model with these considerations in mind, we use the even lower value of $\mu = 14$ km, which we found to result in a realistic settling altitude for the sulfate tracer distribution (see Appendix B).

Observations giving the total injected mass and e-folding time for SO$_2$ (30 days) and ash (1 day) for the Mt. Pinatubo eruption were estimated from satellite data and published in Guo et al. (2004a, b) and Barnes and Hofmann (1997). Table 1 provides the chosen parameter values. In particular, the model describes a 24-hour injection of a plume centered on 14 km in the vertical, uniformly over a single column. We assume no background values for any of the injected species prior to the eruption, as in some other studies (e.g. Bekki and Pyle (1994)).

The remainder of the model formulation as presented in Sect. 3.2-3.4 is applied uniformly on each timestep, and thus the temporal index $n$ will be omitted for brevity. Optical depths and radiative forcings are computed identically for each tracer



species, and so the tracer index $j$ will be also be omitted. Mixtures of multiple tracers $j$ with varying radiative extinction coefficients will be reintroduced in Sect. 3.4.3.

**Table 1.** Model parameters. Parameters with a superscript † are tuned parameters. Parameters with a superscript ‡ are constrained by a data-driven calculation, though not necessarily free for tuning. Parameters without a superscript are observations and/or estimates directly from the literature. For information on tuning, see Sect. 3.5 and Appendix B.

| Parameter | Value | Units | Description | Reference |
|---|---|---|---|---|
| **injection parameters** | | | | |
| $\phi_0$ | 15.15 | deg | meridional plume center | |
| $\lambda_0$ | 120.35 | deg | zonal plume center | |
| $\delta t$ | 24 | hr | injection duration | |
| $\mu$ | 14 | km | peak injection altitude | Stenchikov et al. (2021) |
| **tracer parameters** | | | | |
| $k_{SO2}$ | 1/25 | $day^{-1}$ | SO$_2$ decay rate | Guo et al. (2004a) |
| $k_{sulfate}$ | 1/360 | $day^{-1}$ | sulfate decay rate | Barnes and Hofmann (1997) |
| $k_{ash}$ | 1 | $day^{-1}$ | ash decay rate | Guo et al. (2004b) |
| $M_{SO2}$ | 17 | Tg | injected mass of SO2 | Guo et al. (2004a) |
| $M_{ash}$ | 50 | Tg | injected mass of ash | Guo et al. (2004b) |
| $\nu^{\ddagger}$ | 2.04 | - | SO$_2 \rightarrow$ sulfate weighting | See Sect. 3.2 |
| **heating parameters** | | | | |
| $\zeta^{\dagger}$ | $4.0 \times 10^{-3}$ | - | surface heat transfer efficiency | See Sect. 3.4 |
| $\widetilde{\delta z}^{\dagger}$ | 100 | m | max height of surface cooling | See Sect. 3.4 |
| $b_{SW, ash}^{\ddagger}$ | 400 | $m^2\,kg^{-1}$ | SW mass extinction coeff. | See Sect. 3.3 |
| $b_{SW, SO_2}^{\ddagger}$ | 400 | $m^2\,kg^{-1}$ | SW mass extinction coeff. | See Sect. 3.3 |
| $b_{SW, sulfate}^{\ddagger}$ | 1900 | $m^2\,kg^{-1}$ | SW mass extinction coeff. | See Sect. 3.3 |
| $b_{LW, ash}^{\dagger}$ | $1 \times 10^{-5}$ | $m^2\,kg^{-1}$ | LW mass extinction coeff. | See Sect. 3.3 |
| $b_{LW, SO_2}^{\dagger}$ | 0.01 | $m^2\,kg^{-1}$ | LW mass extinction coeff. | See Sect. 3.3 |
| $b_{LW, sulfate}^{\dagger}$ | 29 | $m^2\,kg^{-1}$ | LW mass extinction coeff. | See Sect. 3.3 |

## 3.2 Sulfate formation

Once injected into the atmosphere, SO$_2$ follows an oxidation chain with an end product of sulfuric acid (H$_2$SO$_4$) that condenses with water vapor to form sulfate aerosol particles (Bekki, 1995). Stratospheric sulfate aerosols have an e-folding removal



timescale of one year (Barnes and Hofmann, 1997), and are responsible for much of the heating that perturbs the Earth's energy balance and atmospheric circulation after a stratospheric volcanic eruption (McCormick et al., 1995; Robock, 2002).

In fully coupled climate models, aerosol heating will be mediated by chemistry, radiation, and moist subgrid processes. Here, the same heating is rather modeled by a direct, analytic coupling of $SO_2$ to sulfate, in a way inspired by the so-called "toy chemistry" of Lauritzen et al. (2015), also seen in Toohey et al. (2016). Sulfate will evolve by Eq. (6), where the source term $f$ exactly becomes the $SO_2$ sink $R(m_{SO2})$. The $SO_2$ removal rate $k_{SO2}$ is then interpreted purely as a reaction rate, and the sulfate tendency mass is

$$\frac{\partial m_{\text{sulf}}}{\partial t} = -k_{\text{sulf}}\, m_{\text{sulf}} + \nu\, k_{\text{SO2}}\, m_{\text{SO2}} \tag{16}$$

or, in terms of mixing ratio on the computational grid,

$$\frac{\partial q_{\text{sulf},i,k}}{\partial t} = -k_{\text{sulf}}\, q_{\text{sulf},i,k} + \nu\, k_{\text{SO2}}\, q_{\text{SO2},i,k} \;. \tag{17}$$

Here, the reaction weight $\nu$ encodes the net production of sulfate per unit mass of $SO_2$. While $\nu$ could be a tuning parameter of the model, we can inform a first choice from chemistry. Since the overall effect of the oxidation sequence yields one aerosol "particle" of sulfate per molecule of $SO_2$ (Bekki, 1995), $\nu$ will just be the ratio of the sulfate to $SO_2$ molar mass. Though it is

known from observation that sulfate particles vary in their composition across latitude, altitude, and season (Yue et al., 1994), depending on specific humidity and temperature, we make the simplifying assumption that all sulfate particles are 75% $H_2SO_4$ by mass. The same assumption is made in Bekki (1995) and suggested by observation (Rosen, 1971; Yue et al., 1994). Defining this percentage as $f_{\text{acid}} = 0.75$, and the molar masses of $H_2SO_4$ and $SO_2$ as $M(H_2SO_4)$ and $M(SO_2)$, the reaction weighting is

$$\nu = \frac{M(H_2SO_4)/f_{\text{acid}}}{M(SO_2)} \approx \frac{1/0.75 \times 98.079 \text{ g/mol}}{64.066 \text{ g/mol}} = 2.04 \;. \tag{18}$$

This choice of $\nu$ results in a peak sulfate mass of about $\sim$28 Mt occurring approximately two months after injection, which is consistent with previous modeling efforts by e.g. Bluth et al. (1997). In that study, however, the authors note that if we assume sulfate production to arise directly from $SO_2$ depletion, then the inferred sulfate loading does not coincide with observed 0.55 $\mu$m AOD anomalies after Pinatubo. Citing the AOD database of (Sato et al., 1993), Bluth et al. (1997) show this peak AOD

anomaly to occur nearer to nine months than two months.

For this reason, Toohey et al. (2016) (who also modeled the $SO_2 \rightarrow$ sulfate conversion directly), decided to address this lag in AOD anomaly by artificially inflating the $SO_2$ dissipation parameter to $k_{SO2} = 1/180 \text{ day}^{-1}$. This change is said to represent the *net* timescale of all processes resulting in increased global mean AOD, beyond just the oxidation chain producing $H_2SO_4$, which may not be fully captured in this idealized description. In this way, they delay the peak sulfate loading, and thus peak

0.55 $\mu$m AOD anomaly, from two months to six months post-injection. This figure is more consistent with the Pinatubo AOD anomaly time series constructed by the Chemistry-Climate Model Initiative (CCMI; Eyring et al. (2013)).

Rather than following these findings of Bluth et al. (1997) and Toohey et al. (2016), we decide instead to retain the observed value of $k_{SO2} = 1/30 \text{ day}^{-1}$. This choice causes the peak AOD anomaly to occur simultaneously with the sulfate loading near





month two, which is consistent with 0.55 $\mu$m AOD results of the prognostic aerosol implementation of Brown et al. (2024),

as well as observations of 0.6 $\mu$m AOD from the Advanced Very High Resolution Radiometer (AVHRR; Zhao et al. (2013);

Heidinger et al. (2014)). We verified that the difference in peak sulfate mass between our model and Toohey et al. (2016) is

explained fully by the choice of $k_{\text{SO2}}$, and not the reaction normalization $\nu$.

### 3.3 Aerosol optical depth

A single aerosol species can contribute to extinction of transmitted radiation by either absorption and scattering, the combined

effect of which is expressed by a spatially-varying extinction coefficient $\beta_e(x, y, z)$. Within a single model column, we will

make the parallel plane approximation, i.e. $\beta_e(x, y, z) \approx \beta_e(z)$. This coefficient can further be expanded as

$$\beta_e = b_e \, \rho = b_e \, q \, \rho_{atm}, \tag{19}$$

where $b_e$ is the *mass extinction coefficient* of the aerosol species, with dimensions of area per unit mass, $\rho$ is the tracer mass

density, and $q$ is the mixing ratio. Consistent with Sect. 2.3, the extinction properties of each tracer species $j$ will be modeled

with respect to two broadbands: $b_{\text{LW}}$ will be used for the extinction of longwave radiation, which is assumed to be entirely

absorption, and $b_{\text{SW}}$ will be used for the extinction of shortwave radiation, which is assumed to be entirely scattering:

$$b_{\text{LW}} \equiv (b_e \text{ for the longwave band}),$$

$$b_{\text{SW}} \equiv (b_e \text{ for the shortwave band}).$$

The remainder of this subsection will only discuss $b_{\text{SW}}$. The longwave extinction $b_{\text{LW}}$ will be used in Sect. 3.4.1 for the local

heating by longwave absorption, but does not contribute to the total column AOD.

For a column with a model top at $z_{\text{top}}$, the dimensionless AOD $\tau$ at a height $z$ is obtained by vertically integrating the

shortwave extinction:

$$\tau(z) = \int\limits_{z}^{z_{\text{top}}} \beta_e(z') \, dz' = \int\limits_{z}^{z_{\text{top}}} b_{\text{SW}} \, q(z') \, \rho_{atm}(z') \, dz' \tag{20}$$

On the model grid, this extinction becomes

$$\tau_{i,k} = \sum_{k' < k} b_{\text{SW}} \, q_{i,k'} \, \rho_{\text{atm},i,k'} \, \Delta z_{i,k'} \tag{21}$$

$$= \sum_{k' < k} b_{\text{SW}} \frac{q_{i,k'} \, \Delta p_{i,k'}}{g}. \tag{22}$$

where the pressure thickness $\Delta p$ symbolizes the pressure difference between two neighboring model interface levels that

surround the full model level with index $k'$. We assume that the indices $k$ and $k'$ decrease toward the model top (as in E3SMv2).

We also define a shorthand for the cumulative AOD at the surface as $\tau_i \equiv \tau(z = 0)$. After summing over $k$ for this case, we

have the usual result (Petty, 2006) that each remaining term is just the total column mass burden $M_i$ of the tracer, scaled by the

mass extinction coefficient $b_{\text{SW}}$ and column area $a_i$,

$$\tau_i = \sum_k b_{\text{SW}} \frac{q_{i,k} \Delta p_{i,k}}{g} = \sum_k b_{\text{SW}} \frac{q_{i,k} m_{atm,i,k}}{a_i} = b_{\text{SW}} \frac{M_i}{a_i}. \tag{23}$$



### 3.4 Radiative forcing

Injected stratospheric aerosols force the Earth system in two primary ways which are (1) local heating of the stratosphere and
(2) remote cooling of the surface. The presence of $SO_2$ and sulfate aerosols in the stratosphere induces a local diabatic heating
to the temperature field by absorption of upward-propagating longwave radiation (Kinne et al., 1992; Brown et al., 2024). After
the 1991 Mt. Pinatubo eruption, this process resulted in a positive temperature anomaly of up to $\sim$2–4 K peaking near 50–30
hPa (Rieger et al., 2020; Stenchikov et al., 1998; Labitzke and McCormick, 1992), driven by a maximum net temperature
change at a rate of $\sim$1 K month$^{-1}$ during the initial period following the injection.

At the same time, increased aerosol optical depths of the vertical column decrease the flux density of shortwave solar
radiation reaching the troposphere. This upper-level scattering of solar radiation contributed to an observed surface cooling of
$\sim -0.5$ K during the two years following the eruption of Mt. Pinatubo (Dutton and Christy, 1992; Self et al., 1993; Fyfe et al.,
2013).

We model each of these heating effects by adding new forcing terms to the temperature field of the HSW atmosphere.
Heating is applied in the stratospheric aerosol plume, and the lowest few model levels are cooled, via the computation of the
energy change that results from the attenuation of the flux densities $I_{LW}$ and $I_{SW}$.

### 3.4.1 Local heating of the stratosphere

The local warming effect is modeled as an attenuation of upwelling longwave radiation with flux density $I_{LW}$ defined in
Eq. (4), computed for each model column via the plane-parallel Beer-Lambert law. To begin, the attenuated flux density after
transmission through a particular slab with vertical bounds $[z_0, z_1]$ is an integral of the extinction $\beta_e$ such as

$$I(z_0, z_1) = I_{LW} \exp\left(-\int_{z_0}^{z_1} \beta_e(z')\, dz'\right). \tag{24}$$

Here we assume that $z_0$ is the lowest extent of the aerosol plume, and there has been no attenuation between $z = 0$ and $z = z_0$.
In this case, the power per unit area absorbed by the slab is

$$\Delta I = I_{LW} - I(z_0, z_1). \tag{25}$$

If we consider another slab located immediately above $z_1$, on $[z_1, z_2]$, then the incident flux is no longer $I_{LW}$, but rather
$I(z_0, z_1)$, and the power per unit area absorbed is

$$\begin{aligned}
\Delta I &= I(z_0, z_1) - I(z_1, z_2) \\
&= I_{LW} \exp\left(-\int_{z_0}^{z_1} \beta_e(z')dz'\right)\left[1 - \exp\left(-\int_{z_1}^{z_2} \beta_e(z')dz'\right)\right].
\end{aligned} \tag{26}$$



This form generalizes to an arbitrary slab on $[z_n, z_{n+1}]$ as

$$\Delta I = I(z_{n-1}, z_n) - I(z_n, z_{n+1})$$

$$= I_{\mathrm{LW}} \exp\left( -\int_{z_0}^{z_n} \beta_e(z') dz' \right) \left[ 1 - \exp\left( -\int_{z_n}^{z_{(n+1)}} \beta_e(z') dz' \right) \right]. \tag{27}$$

Discretizing these integrals onto the vertical grid with levels $k$ and column $i$ yields

$$\Delta I_{i,k} = I_{\mathrm{LW}} \exp\left( -\sum_{k'>k} b_{\mathrm{LW}} \frac{q_{i,k'} \Delta p_{i,k'}}{g} \right) \left[ 1 - \exp\left( -b_{\mathrm{LW}} \frac{q_{i,k} \Delta p_{i,k}}{g} \right) \right], \tag{28}$$

where the argument to the leftmost exponent sums over all levels $k'$ which are below level $k$. The effect here is that aerosols lower in the vertical column "shadow" those above, decreasing the power of incident radiation available for absorption. In this way, the peak of the local aerosol heating may lie below the actual density peak of the plume.

The absorbed power per unit area is then translated to a heating rate per unit mass $s$, and finally to an associated temperature tendency $\Delta T$, with the assumption that all of the absorbed radiation is perfectly converted to heat. If the flux densities are given in units of W m$^{-2}$, then by dimensional analysis

$$s_{i,k} = \frac{a_i \Delta I_{i,k}}{m_{i,k}} \frac{\mathrm{J}}{\mathrm{kg\, s}} \tag{29}$$

$$\implies \Delta T_{i,k} = \frac{1}{c_p} \frac{a_i \Delta I_{i,k}}{m_{i,k}} \frac{\mathrm{K}}{\mathrm{s}}. \tag{30}$$

This temperature tendency is always positive, and will be imposed on the grid cell at $(i, k)$ for each tracer at each timestep.

### 3.4.2 Cooling of the surface

The surface cooling is modeled as an AOD attenuation of incident radiation with flux density $I_{\mathrm{SW}}$, as defined in Eq. (5). We begin with an analogous form to Eq. (24), where the vertical slab on $[z_0, z_1]$ is replaced with the entire vertical column above position $z$, on $[z, z_{\mathrm{top}}]$. The integral term in brackets is then exactly the AOD as given in Eq. (20). The attenuation is thus

$$I(z) = I_{\mathrm{SW}} \exp\left( -\int_{z}^{z_{\mathrm{top}}} \beta_e(z') dz' \right) = I_{\mathrm{SW}}\, e^{-\tau(z)}. \tag{31}$$

With the notation used in Eq. (23), the deficit flux density after attenuation by the aerosol over the full height of the atmosphere on a single model column $i$ is

$$\Delta I_i = I_{\mathrm{SW}} \left( e^{-\tau_i} - 1 \right). \tag{32}$$

That is, a deficit energy density of $\Delta I_i$ W m$^{-2}$ is imposed at the surface. For a model column at the equator with $\tau = 0.2$, this form gives $\Delta I \approx -100$ W m$^{-2}$, which is roughly consistent with the observed broadband solar transmission deficits of $\sim 20\%$ at Mauna Loa, Hawaii in the months following Pinatubo (Self et al. (1993); see their Fig. 9). By AODs of $\tau \approx 4$, the shortwave attenuation saturates (all available incident radiation has scattered).





The attenuation is next translated to a cooling rate per unit mass $s$, and an associated temperature tendency $\Delta T$. Since the HSW atmosphere simulates no land-atmosphere coupling processes, we employ a very simple representation of the conduction and convection that would, in reality, be responsible for communicating an energy deficit at the ground to the atmospheric surface layer. We imagine that all of the energy lost over the column heats the planetary surface, which in turn transfers heat to the atmosphere by a function $F$ with some efficiency $\zeta$:

$$s_{i,k} = \zeta F(\Delta I_i). \tag{33}$$

The heat transfer "efficiency" $\zeta$ should be considered a catch-all for any surface-atmosphere coupling effects which we do not model, and is treated as a tuning parameter for the magnitude of atmospheric surface cooling (see Sect. 3.5). As in the local heating treatment of the previous section, the function $F$ can be obtained by dimensional analysis:

$$s_{i,k} = \zeta \frac{a_i \Delta I_i}{\widetilde{m}_i} \frac{\text{J}}{\text{kg s}} \tag{34}$$

$$\implies \Delta T_{i,k} = \zeta \frac{1}{c_p} \frac{a_i \Delta I_i}{\widetilde{m}_i} \frac{\text{K}}{\text{s}}, \tag{35}$$

where $\widetilde{m}_i$ is the mass of air in the lowest $\kappa$ model levels of column $i$ over which the cooling is to be applied. If we apply the cooling only to the lowest model level with $\kappa = K$, then

$$\widetilde{m}_i = m_{i,K} \tag{36}$$

and otherwise

$$\widetilde{m}_i = \sum_{k=K}^{K-(\kappa-1)} m_{i,k}. \tag{37}$$

In this way, the net cooling (total energy loss over unit time) is conserved as the parameter $\kappa$ is increased, and the cooling per unit mass is "diluted". The choice of $\kappa$ will effectively encode whatever missing physical mechanisms would otherwise communicate the cooling higher into the vertical column. For $\kappa > 1$, $\Delta T_{i,k}$ is a 3D quantity, while $\Delta I_i$ and $\tau_i$ are always 2D quantities. In Table 1, rather than setting $\kappa$ directly we set $\widetilde{\delta z}$, or the height above the surface in meters where the cooling

should be applied, from which $\kappa$ is inferred, given the vertical discretizaiton.

### 3.4.3    Generalization to mixtures of tracer species

When multiple tracer species $j$ are present (SO$_2$, ash, sulfate), the total radiative heating is not derived from a simple sum of the $\Delta T$ solutions found over the proceeding sections. Rather, it is the total extinction which is determined by additive extinction coefficients,

$$\beta_e = \sum_j \beta_{e,j} = \sum_j b_{e,j} m_j. \tag{38}$$





In this case, the total AOD of Eq. (23) becomes

$$\tau_i = \sum_k \sum_j b_{\text{SW},j} \frac{q_{j,i,k} \Delta p_{i,k}}{g} = \sum_j b_{\text{SW},j} \frac{M_{j,i}}{a_i} = \sum_j \tau_{j,i}. \tag{39}$$

For the total longwave heating, the expression is somewhat more complicated. Equation (28) becomes

$$\Delta I_{i,k} = I_{\text{LW}} \exp\left(-\sum_j \sum_{k'>k} b_{\text{LW},j} \frac{q_{j,i,k'} \Delta p_{i,k'}}{g}\right) \left[1 - \exp\left(-\sum_j b_{\text{LW},j} \frac{q_{j,i,k} \Delta p_{i,k}}{g}\right)\right]. \tag{40}$$

Here, each grid cell has an incident flux density that has already been attenuated by *all* species $j$ underneath it, and so the total attenuation is not simply a sum of $j$ separate evaluations of $\Delta I$.

## 3.5  Model summary & parameter tuning

Figure 3 provides a summary of the important equations developed in the previous subsections, and Table 1 gives the chosen parameter values. Some parameter values are taken directly from observations or previous works in the literature, while others
are derived quantities. Five parameters are tuning parameters, including the longwave mass extinction coefficients for $SO_2$, sulfate, and ash, the maximum height of forced surface cooling, and the surface heat transfer efficiency.

The longwave attenuation mechanism of the model is tuned to produce realistic stratospheric heating rates by sulfate aerosols. The mass extinction coefficient $b_{\text{LW}}$ for sulfate is instrumental in tuning the long-term mean temperature anomalies. Not as obvious is the importance of $b_{\text{LW}}$ for the very short-lived ash tracer. The lofting speed of the plume will be controlled by
the aggressive early heating of ash in the fresh plume (Stenchikov et al., 2021), since the initial ash mass loading (50 Tg) is dominant over that of $SO_2$ (17 Tg). As such, the mass extinction coefficient for ash serves as the main tuning parameter which controls the settling height of the aged aerosols. $SO_2$, on the other hand, participates both in the initial lofting of the plume, as well as the short-term temperature anomalies for the first couple months.

The shortwave mass extinction coefficients $b_{\text{SW}}$ do not play the same role in tuning the surface cooling. Instead, we simply
constrain $b_{\text{SW}}$ of each species to yield an AOD representative of post-Pinatubo zonal-mean observations. During the months and years following the eruption, these values peaked near 0.2-0.5 (Toohey et al., 2016; Mills et al., 2016; Stenchikov et al., 2021; Dutton and Christy, 1992; Stenchikov et al., 1998). Tuning the magnitude of surface cooling is then passed on to the efficiency parameter $\zeta$.

A description of the actual tuning process, as well as recommendations for tuning the model on different simulation grids
and varying aerosol injection scenarios can be found in Appendix B.





---

**Tracer tendencies**

$$\frac{\partial q_{j,i,k,n}}{\partial t} = \frac{g}{\Delta p_{i,k} a_i} \left[ -k_j m_{j,i,k} + I_{i,n} \frac{M_j}{\delta t \sum_k V_k} V_k \right] \quad (14)$$

$$\frac{\partial q_{\text{sulf},i,k,n}}{\partial t} = -k_{\text{sulf}} q_{\text{sulf},i,k,n} + w k_{\text{SO2}} q_{\text{SO2},i,k,n} \quad (17)$$

$$V(z) = \exp\left( -\frac{1}{2} \frac{(z-\mu)^2}{(1.5 \text{ km})^2} \right) \quad (15)$$

**LW radiative and optical properties**

$$I_{\text{LW}} = \sigma \left[ 315\text{K} - (60\text{K})\sin^2\phi \right]^4 \quad (4)$$

$$\Delta I_{i,k} = I_{\text{LW}} \exp\left( -\sum_j \sum_{k'>k} b_{\text{LW},j} \frac{q_{j,i,k'} \Delta p_{i,k'}}{g} \right) \left[ 1 - \exp\left( -\sum_j b_{\text{LW},j} \frac{q_{j,i,k} \Delta p_{i,k}}{g} \right) \right] \quad (40)$$

$$s_{i,k} = \frac{a_i \Delta I_{i,k}}{m_{i,k}} \frac{\text{J}}{\text{kg s}} \quad (29)$$

**SW radiative and optical properties**

$$I_{\text{SW}} = I_0 \cos\phi \frac{\text{W}}{\text{m}^2} \quad (5)$$

$$\tau_i = \sum_j b_{\text{SW},j} \frac{M_{j,i}}{a_i} \quad (39)$$

$$\Delta I_i = I_{\text{SW}} \left( e^{-\tau_i} - 1 \right) \frac{\text{W}}{\text{m}^2} \quad (32)$$

$$s_{i,k} = \zeta \frac{a_i \Delta I_i}{m_i^{\text{cool}}} \frac{\text{J}}{\text{kg s}} \quad (34)$$

**Figure 3.** Summary of the important model equations controlling the tracer injection and removal, and radiative and optical properties for the tracers in shortwave and longwave broadbands. See equation numbers in the text for explanations. The SW and LW equations are written for a mixture of tracer species $j$ at a fixed timestep $n$. Values for the parameters are given in Table 1.

## 4 Implementation in E3SMv2

### 4.1 Ensemble generation

We explored two different ensemble generation strategies, which we are characterized by a "high variability" (HV), or a "limited variability" (LV) set of initial conditions. Both strategies appear in the literature, though not often explicitly named and compared.

In the HV strategy, ensemble member initial conditions are sampled from a base run of the HSW climate (described in Sect. 2) at an interval which produces *independent* atmospheric states. The choice of this time interval is unique to the model configuration. In making this determination, we follow the methodology of Gerber et al. (2008). In short, an index measuring the dynamical process which sets the upper-bound on low-frequency variability in the model is defined, and the time that it takes for the autocorrelation of this index to vanish is found. At that time, we consider the initial condition to have been "forgotten". For the HSW forcing, no seasonal cycle or ocean process are imposed, and so the upper-bound variability timescale is set by positional variations of the extratropical jets, encoded as the annular mode index (defined in Gerber et al. (2008)).

For a standard HS94 forcing on a ~1-degree pseudospectral grid, Gerber et al. (2008) showed the annular mode index autocorrelation to vanish by day 90–100. Lower resolution grids had progressively longer timescales. We found that this autocorrelation was ~0.1 by day 90 for the HSW atmosphere on the ne16pg2 grid in E3SMv2, thereafter only slowly converging to zero by day ~250. Compromising on these diminishing returns for efficiency, our HV ensembles are generated by sam-





pling initial states from a base run every 90 days. Volcanic injections can then begin at any point in the individual member integrations.

In the LV strategy, all ensemble members are initialized with an identical state, which is subjected to a random gridpoint-level temperature perturbation of $1 \times 10^{-4}$ K. We then wait some amount of time before enabling the volcanic injections. During this pre-injection period, the members will diverge from one another as dynamical feedbacks seeded by the initial temperature perturbations grow. In our experiments, waiting 75 days produced ensemble member background states that are more qualitatively similar in their zonally-averaged flow, but exhibit synoptic-scale variations.

Note that the two timescales quoted above in the generation of the HV and LV ensembles are distinct, and should not be confused. In the former case, the initial conditions lie 90 days apart from one another, while in the latter case, the perturbed initial conditions evolve together, though slowly diverge, for a period of 75 days.

With enough members, the HV ensemble mean will show the average atmospheric response to our volcanic forcing *independent* of the background state. Meanwhile, an LV ensemble mean will show the robust response to a *particular* state, at least for the initial plume evolution. Eventually, the LV members will diverge, and will be statistically similar to the HV ensemble once the aerosol distribution approaches zonal symmetry. Thus, an LV ensemble is perhaps most interesting to studies of this early phase. In Sect. 4.2, only show results from an HV ensemble, though we encourage future experiments to present their ensemble generation methods in these terms.

Figure 4 shows a snapshot of a five-member volcanic injection ensemble at eight-days post-injection for the HV and LV ensemble generation strategies. The differences seen here are principally due to the fact that the HV ensemble samples strongly varying states of the northern polar jet, while the bulk aerosol transport of the plumes of the LV ensemble follow each other more closely. Also shown are time histories of the averaged zonal-mean zonal wind within 20 degrees in latitude of the eruption site (from 5°S to 35°N), at 50 hPa for all ensemble members, demonstrating the difference between the background HV and LV states.

Figure 5 displays the zonal-mean of the initial conditions for temperature and zonal wind for ensemble members ens01, ens03, and ens05 for the HV case. These states show the qualitative spread in independent states sampled from an evolving HSW atmosphere, the most notable differences being the balance between (or collapse of) the polar jets, and the strength of the stratospheric equatorial easterlies. Note that the zero-contour rises steeply from the tropical to midlatitude region, and thus the initial transport of a plume for a fixed height will vary strongly with latitude, and will also be particularly sensitive to movements of the jet stream. The combination of the chosen initial condition and the parameter configuration given in Table 1 results in the lower-tail of the initial injection distributions catching westerlies, while most of the mass enters the stratosphere and travels East. Note that all of the initial conditions for the LV case were based on perturbations of the HV ensemble member "ens05" (Fig. 5 panel (c)).

## 4.2 Discussion of the results

We ran a five-member HV ensemble (Sect. 4.1) of E3SMv2 simulations subject to the modified idealized HSW physics (Sect. 2, Appendix A). A volcanic injection of $SO_2$ and ash (Sect. 3) occurs at day 180, with the parameter configuration of Table 1.



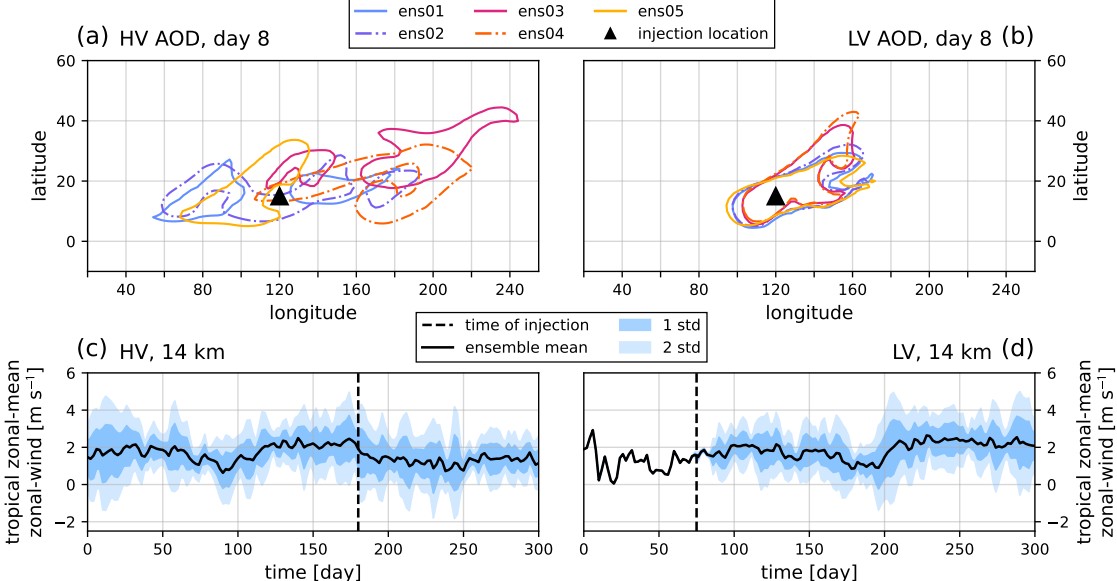

**Figure 4. (a)** AOD 0.5 contours for five HV ensemble members at eight-days post-injection. Each ensemble member is given a unique color, and their line styles alternate for visual clarity. **(b)** Identical to panel (a), but for five LV ensemble members. **(c)** Zonal-mean zonal wind averaged over a tropical region bracketing the injection site, from 5°S to 30°N, at 50 hPa, for the HV ensemble. A bold black line shows the ensemble mean. Dark and light blue shading show one and two standard deviations, respectively. A black vertical dashed line shows the time of injection (day 180). **(d)** Identical to panel (c), but for the LV ensemble, with injection at day 75.

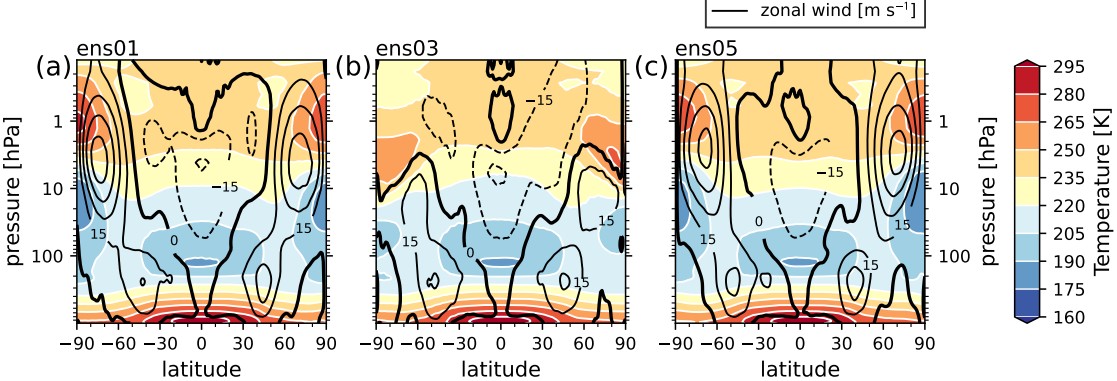

**Figure 5.** Zonal-mean of the initial condition for ensemble members ens01, ens03, and ens05 for the HV ensemble (corresponding to the solid-line AOD distributions of Fig. 4). Temperature is shown on the color scale with intervals of 15 K, and zonal wind in black contours with intervals of 15 m s$^{-1}$. The zero m s$^{-1}$ contour in zonal wind is shown in bold, and negative contours are dashed.





Figure 6 shows the transport of the $SO_2$ and sulfate aerosol plumes at the 45 hPa model level for days 10, 20, 40, and 80 for the single ensemble member "ens01". At this altitude, the dominant transport is driven by the easterly winds of the tropical stratosphere (see Fig. 1). By day 20 the plume has circled the globe, and by day 40 the plume has reached the northern pole. Also during this time, both $SO_2$ and sulfate concentrations have risen for this fixed vertical level. For $SO_2$, this effect is
purely driven by vertical transport (our model contains no gravitational settling of any tracer species, and so all species will dynamically loft as long as heating is present), while for sulfate, this effect is a combination of transport and actual aerosol production. By day 80, the tracer distributions are well-mixed in the tropical and midlatitude regions, and increasingly more $SO_2$ has been converted to sulfate.

Figure 7 provides a detailed view of the ash plume evolution over the first 20 days of the simulation. Panel (a) shows the
zonal-mean ash mixing ratios as a function of time and pressure, averaged over a 20° band centered on the injection in latitude, from 5°S to 35°N. By day 12, the zonal-mean ash mixing ratios in this region have dissipated below $10^{-12}$. Also shown for reference is the growing sulfate plume, which is just starting to be produced by $SO_2$ conversion. Panel (b) shows the total amount of ash removed from the stratosphere over the same time period, in g m$^{-2}$. That is, we are plotting the cumulative sum of the removal function $R(m_{ash})$ (Eq. (7)) over all grid cells above 100 hPa, from days 180 through 200. Our model does
not actually implement gravitational settling of ash, though our simple removal process can be thought of as an accumulated "fallout". Thus, this distribution shows both the extent and history of the ash plume after 20 days.

Figure 8 displays the evolution of the zonal-mean AOD as a function of latitude and time, as well as the imposed radiative cooling rate at the surface in (K day$^{-1}$) by SW extinction. The AOD peaks at 0.3 near 15°N after one month, and by day 90 post-injection, zonal-mean optical depths of 0.1 reach the northern pole. Figure 9 shows the zonal-mean distribution of sulfate,
and the local stratospheric heating rate by LW absorption, as a 30-day time average over days 60 through 90 post-injection. The aerosol density and heating rates coincide with one another in the tropics, while at higher latitudes, the heating rate distribution develops strong meridional gradients that the sulfate mixing ratio does not. These gradients are an imprint of the LW radiation profile of Fig. 2, which is minimized at the poles.

Several features of the HSW general circulation are exhibited in Figs. 8,9. A realistic tropopause is formed by the inversion
of the equilibrium temperature $T_{eq}$ near 130 hPa in the tropics. At the same time, the shape of $T_{eq}$ at the lowest model levels mimics unequal solar insolation of the surface, driving convection in the tropical troposphere. Together, these effects give rise to upper-level divergence at the tropical tropopause, and subsidence in the subtropics. The resulting Hadley cell can be seen in the sulfate distribution tail descending to the surface south of 30°N. The meridional transport of the zonally-averaged tracer distribution, however, appears not to be hemispherically symmetric, with most of the aerosol population remaining in
the northern hemisphere.

For the Mt. Pinatubo eruption, relative hemispheric symmetry of the AOD and temperature signal is established much more rapidly both in observations (Stenchikov et al., 1998; Mills et al., 2016), and in more realistic models (Mills et al., 2016; Stenchikov et al., 2021; Ramachandran et al., 2000; Karpechko et al., 2010; Brown et al., 2024). This hemispheric symmetry is imposed because, in reality, the mean meridional circulation is characterized during solstice months by a strong
winter hemisphere Hadley cell, a relatively weak cell in the summer hemisphere, and a convergence zone north of the equator





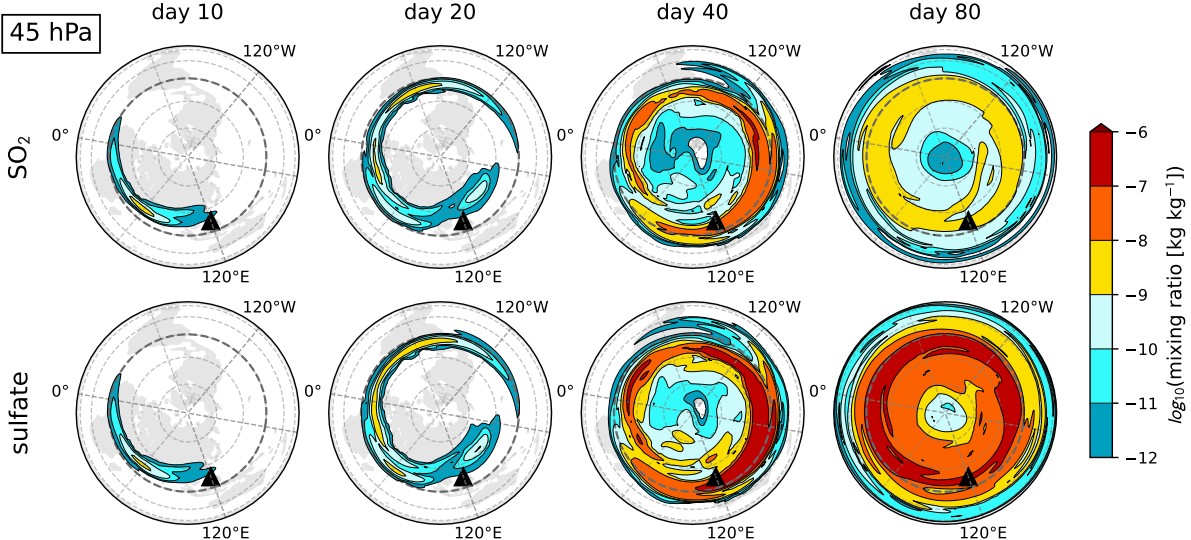

**Figure 6.** SO$_2$ (top row) and sulfate aerosol (bottom row) mixing ratios in (kg tracer)(kg dry air)$^{-1}$ for a single ensemble member at the 45.67 hPa model level, displayed with a logarithmic scale. Columns from left-to-right correspond to 10, 20, 40, and 80 days post-injection. The data is plotted on a Lambert azimuthal equal-area projection extending from the north pole to 60°S, where continental landmasses are shown only for spatial reference (our model features no topography or land processes). A 30°×30° grid is drawn in dashed lines, with the equator in bold dash. The injection location is marked with a black triangle.

(Schneider et al., 2014), driven by the seasonal cycle (Schneider, 2006), as well as asymmetry in the northern and southern land-sea distribution (Cook, 2003). During the northern hemisphere summer of July 1991, the upper-level diverging branch of the southern cell would have readily facilitated cross-equatorial transport of stratospheric aerosols (Hoskins et al., 2020). All of these features are absent from our axisymmetric model, where any air masses in the upper troposphere or above will essentially
always diverge from the equator. To see this flow feature, the HSW Hadley cells are visualized via the Stokes streamfunction $\psi(\phi, p)$ in Fig. 9. Following previous studies (Oort and Yienger, 1996; Cook, 2003; Pikovnik et al., 2022), the $\psi$ function is defined by a vertical integration of the zonally-averaged meridional wind $\bar{v}$ as

$$\psi(\phi, p) = \frac{2\pi a cos\phi}{g} \int\limits_{0}^{p} \bar{v}(\phi, p')dp' \tag{41}$$

where $a$ symbolizes the Earth's radius. At a position in pressure and latitude, $\psi$ gives the mean meridional mass transport over
the entire stratospheric column above. Thus, positive (negative) peaks in the streamfunction distribution indicate clockwise (counterclockwise) circulation in the zonal average. The HSW streamfunction is anti-symmetric about the equator, more closely resembling an equinox state in nature.

Next, we quantify impacts by the volcanic aerosol forcing on the atmospheric state by atmospheric variable anomalies. Anomalies are defined as a the gridpoint-level arithmetic difference between a particular run (or ensemble mean), and the



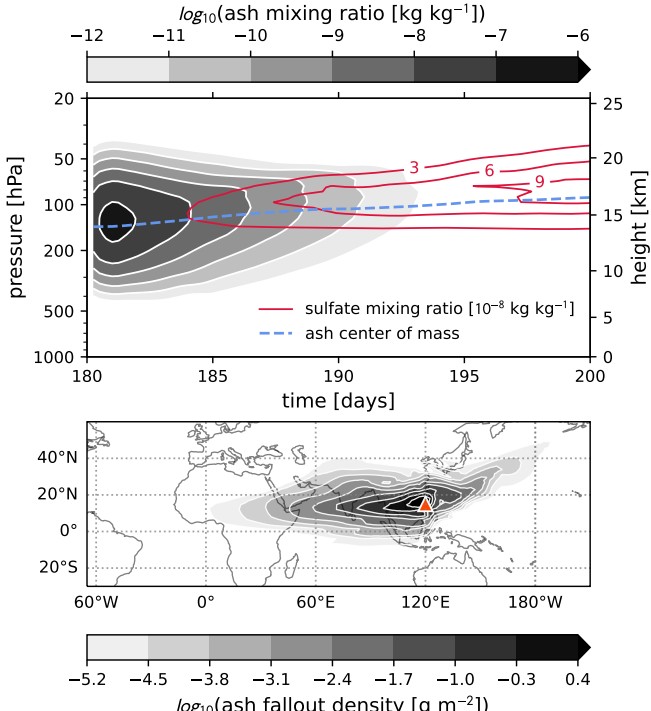

**Figure 7.** Evolution of the ash plume. **(a)** The zonal-mean, ensemble-mean logarithmic ash mixing ratios, averaged over all latitudes within $20°$ of the injection (from $5°$S to $35°$N). A dashed blue line shows the rising center of mass of the ash, and solid red contours show sulfate mixing ratios in intervals of $3 \times 10^8$. Time values are shown as days since eruption. **(b)** The cumulative sum of removed ("fallout") stratospheric ash $R(m_{ash})$ over days 0 through 20 post-injection, and all grid cells above 100 hPa, for a single ensemble member. Values are logarithmically-scaled densities in g m$^{-2}$. A red triangle marks the position of the volcanic injection. Continental landmasses are shown only for spatial reference (our model features no topography or land processes).

time-average of a volcanically-quiescent reference simulation. For this reference run, we use a 10-year run of the spun-up HSW atmosphere with no volcanic forcing, which is shown in Fig. 1, panel (b). Figure 10 shows the resulting ensemble-mean global-mean temperature anomaly as a function of pressure for 1000 days. The temperature anomaly peaks near 2 K at day 120 post-eruption in the stratosphere near 50 hPa, while the surface (lowest model level) anomaly peaks near $-1$ K. Notice that the surface temperature anomalies exhibit much more noise than those in the stratosphere. In particular, we found that the

stratospheric temperature anomaly is positive for any single ensemble members, whereas the negative surface cooling anomaly is often significant (non-zero to at least one standard deviation) only in the ensemble mean.

Also shown in Fig. 10 are the total (globally-integrated) tracer mass time series for SO$_2$ and sulfate, as well as the vertical center-of-mass (COM) of the tropical stratospheric component of the sulfate distribution. The latter is simply defined as a subset of sulfate which remains above the peak of the vertical injection profile at 14 km or $\sim$130 hPa, and within $20°$ of the



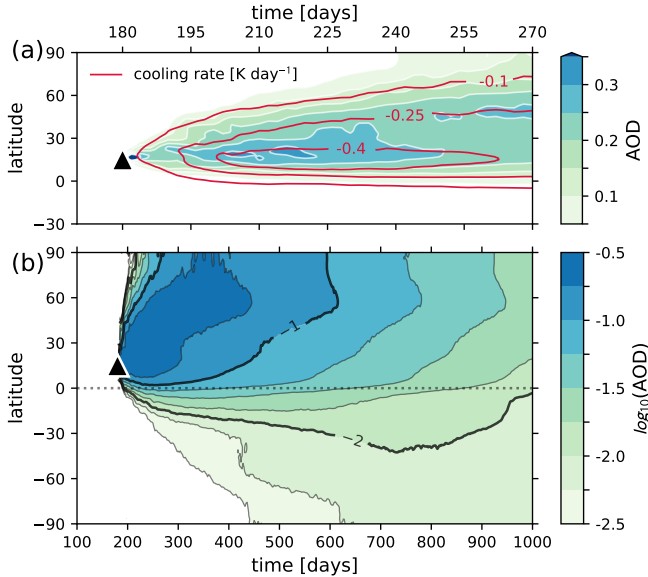

**Figure 8. (a)** Zonal-mean AOD in the latitude-time plane for the first 90 days post-injection. Overplotted is the cooling rate imposed on the lowest model level by shortwave extinction every 0.15 K day$^{-1}$. **(b)** Logarithmic zonal-mean AOD over 1000 days. the 0.1 and 0.001 AOD lines are in bold. A faint dotted line shows the equator, and a black triangle shows the time and vertical center of the injection.

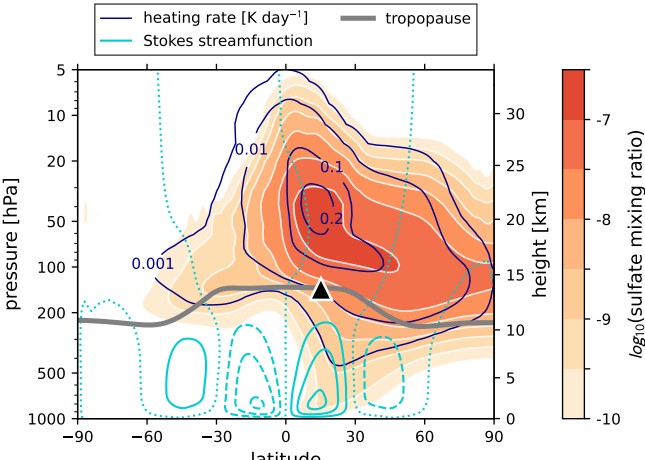

**Figure 9.** 30-day time average over days 60-90 post-injection of the logarithmic zonal-mean sulfate mixing ratio in (kg tracer)(kg dry air)$^{-1}$. Also plotted in solid dark blue contours is the local stratospheric heating rate by longwave absorption in (K day$^{-1}$), with logarithmic intervals between contours 0.001, 0.01, and 0.1, and a final contour drawn at 0.2. Cyan contours show the Stokes streamfunction (Eq. (41)) with intervals of $3 \times 10^{10}$ kg s$^{-1}$. Negative (positive) contours are dashed (solid), and the zero line is dotted. A thick gray line shows the tropopause. Height axis is obtained from the model's diagnostic geopotential height.



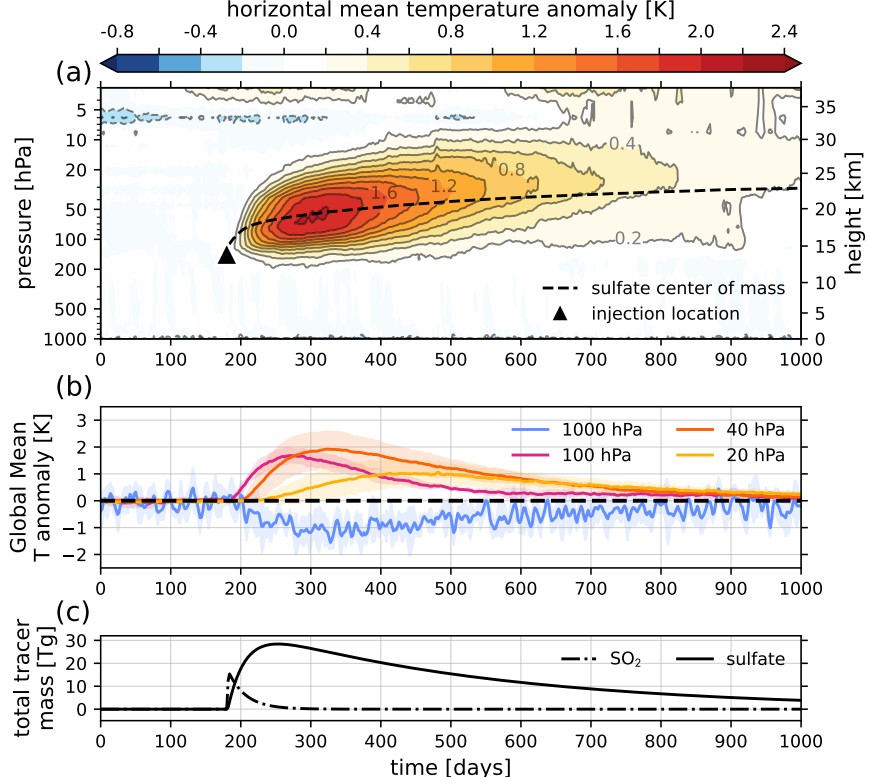

**Figure 10. (a)** Ensemble mean temperature anomalies with respect to a volcanically quiescent reference period of 10 years, averaged at each model level over all longitudes, and all latitudes within 20° of the injection (from 5°S to 35°N). Contour intervals are drawn every 0.2 K. Height axis is obtained from the model's diagnostic geopotential height. Black triangle shows the height of the initial mass injection distribution peak, and time of the eruption at 180 days. A dashed black line shows the center of mass of the volcanic sulfate distribution. **(b)** The temperature anomaly data shown in panel (a), chosen for certain pressure levels, including the surface (1000 hPa). Shading for each curve shows the standard deviation of the ensemble members. **(c)** The globally-integrated tracer mass time series for $SO_2$ and sulfate.

injection in latitude, from 5°S to 35°N. This sulfate subset is the component of the plume most sensitive to radiative heating, and thus largely responsible for the global mean stratospheric temperature anomaly (see Fig. 9).

## 4.3  Computational expense

Activating our scheme in E3SMv2 involves minimal computational overhead. We observed that a ne16pg2 HSW simulation with the volcanic parameterizations turned off runs at 175 simulated model years per wallclock day (SYPD) executing on
384 processes (16 physics columns per process) on the Perlmutter supercomputer at the National Energy Research Scientific Computing Center (NERSC). This is equivalent to 52.6 core-hours (or process-element hours) per simulated year, or "pe-




hrs/yr". With the same specifications, turning on the volcanic parameterizations reduces the throughput to 58.7 pe-hrs/yr, a decrease of ∼10%.

This performance is in contrast to expensive prognostic aerosol implementations in coupled climate models, which often involve modal aerosol size distributions, more detailed radiative bands, and where aerosol interactions involve an inventory of other chemical species which also must be transported. For comparison, Brown et al. (2024) found 5395 pe-hrs/yr executing their prognostic stratospheric aerosol implementation on the higher-resolution ne30pg2 grid in a nudged atmosphere-only configuration of E3SMv2 on the Cheyenne machine at the National Center for Atmospheric Research (NCAR). For a fully-coupled configuration of E3SMv2, Wagman (2024) found 9898 pe-hrs/yr, also on the ne30pg2 grid. Assuming that increasing the resolution from ne16pg2 to ne30pg2 involves a reduction in pe-hrs/yr by a factor of eight our model is ∼11 times faster and ∼21 times faster than atmosphere-only and fully-coupled E3SMv2 simulations, respectively. The assumed factor of eight comes from the fact that when reducing both horizontal dimensions by a factor of two, the timestep must also be halved according to the CFL condition of the dynamical core.

## 5   Conclusions

The injection, evolution, and forcing described in this article constitutes an idealized prognostic simulation of volcanic aerosol emission and impact development. Previously, it has not been possible to include volcanic forcing routines in such a simple environment, as they inherently depend upon a complex library of physical subgrid parameterizations. To our knowledge, there is no other option for simulating sulfur forcing with a prognostic aerosol treatment in a Held-Suarez-based atmosphere.

This idealized prognostic simulation has isolated the volcanic event from other sources of variability, and established a direct relationship between forcing ($SO_2$ emission) and downstream impact (stratospheric and surface temperature anomalies). Delivering these features as a computationally affordable capability facilitates the development of new multi-step data analytic techniques designed to improve downstream attribution. This simulation has been used to develop algorithms to track the space-time evolution of relationships between model fields using a time-dependent directed graph framework (see Bertagna et al. (2024)), and also in the development of explainable AI techniques which measure the importance of input variables on the prediction of downstream temperature (McClernon et al., 2024). We anticipate its broader utility in developing other multi-step attribution methods and in capitalizing on the development of the LV ensemble formulation to establish robust responses to a particular atmospheric state as well.

This work is a new addition to an idealized AGCM model hierarchy that can be used to study phenomena in isolation. Examples of this model hierarchy include Sheshadri et al. (2015) and Hughes and Jablonowski (2023) who studied the effects of topography on the atmospheric flow, or Polvani and Kushner (2002) and Gerber and Polvani (2009) who assessed polar jets and hemispheric asymmetry. Other idealized configurations focus on simple moist flows with moisture feedbacks (Frierson et al., 2006; Thatcher and Jablonowski, 2016), tropical cyclones (Reed and Jablonowski, 2012), tracer-based cloud micro-physics (Frazer and Ming, 2022; Ming and Held, 2018), age-of-air tracers (Gupta et al., 2020), the Madden-Julian oscillation (MacDonald and Ming, 2022), or climate-change forcing (Butler et al., 2010).





We illustrated that our implementation can be used to mimic the spatio-temporal temperature anomaly signatures of large stratovolcano eruptions, and presented one specific parameter tuning that gives rise to a Pinatubo-like event. Our design intentionally leaves out many details which we felt would increase physical complexity, without being necessary for producing realistic atmospheric impacts for attribution studies (e.g. gravitational settling of aerosols). Nevertheless, the formulation remains flexible to modifications. Our parameterizations could support any number of co-injected tracer species, concurrence of multiple eruptions, and injections at any latitude and height. In fact, the description is generic enough that by replacing the vertical and/or temporal injection profiles, we could imagine simulating the aerosol direct-effect of various localized emission events of the troposphere (e.g. wildfire smoke) or the stratosphere (e.g. geoengineering SAI experiments) in an idealized model configuration.

*Code and data availability.* The code for the modified HSW configuration in E3SMv2 and our volcanic emission and forcing parameterizations is available at a public GitHub repository at https://github.com/sandialabs/CLDERA-E3SM. The model code is also available as a tarball on the Zenodo platform, at https://doi.org/10.5281/zenodo.10524801. The majority of our implementation is contained in a single Fortran file, located within the model repository at components/eam/src/physics/cam/cldera_sai_tracers.F90. The subroutines within this module interface with EAM in various places throughout the code, the details of which are beyond the scope of this article. For consultation on setting up and using these parameterizations, users should either contact the corresponding author, or refer to the supplemental namelist settings and model case creation scripts provided in the cited Zenodo repository. Also contained in this Zenodo repository are Python notebooks for reproducing the figures in this manuscript.

## Appendix A:  Modification of the Held-Suarez-Williamson forcing to accommodate higher model tops

As suggested in Sect. 2.2, we make two modifications to the implementation of the HSW forcing scheme which are (1) the adjustment of the radiative equilibrium temperature $T_{eq}$ near the model top, and (2) the inclusion of an additional sponge-layer wind damping mechanism, described in Sect. A1-A2. Figure 1, panel (a) shows the radiative equilibrium temperature with our modifications, and the employed vertical profiles for the sponge layer and surface-layer damping. Figure A1 shows zonally-averaged tropical temperature profiles resulting from five-year E3SMv2 runs responding to the specifications of HS94, W98, and our modified HSW forcings. The three implementations exhibit no difference in tropopause structure. while W98 and our work are consistent until above 2 hPa.

## A1   Modified radiative equilibrium temperature

The model employed in the experiments of W98 features a top at ∼3 hPa, while the standard E3SMv2 model top is located at ∼60 km, or 0.1 hPa. Applying the temperature relaxation profile as published in W98 to E3SMv2 therefore results in undesired reversals in the polar lapse rate near 2 hPa, as well as temperatures at the tropical model top at around 60 km in excess of 300 K. Observed monthly-mean zonal-mean tropical temperatures peaks near 50 km are closer to ∼260 K (Holton and Hakim, 2013). We experimented with modifying the lapse rate parameters in the HSW $T_{eq}$ as suggested by W98 and implemented by





Yao and Jablonowski (2016), but ultimately chose to attempt to retain more realistic temperatures of the upper-stratosphere by simply imposing a lapse rate of zero for $p < 2$ hPa in $T_\text{eq}$. Specifically, the equilibrium temperature used in our modified HSW forcing is

$$T_\text{eq}(\phi, p) = T_\text{eq,HSW}(\phi, \max(p, p^*)) \tag{A1}$$

where $T_\text{eq,HSW}(\phi, p)$ is the form presented in Appendix A of Williamson et al. (1998), and $p^* = 2$ hPa.

Other than this modification, the design and parameter choices of $T_\text{eq}$ are identical to those defined in HS94 and W98. From W98, we inherit the property that the original implementation of HS94 applies below 100 hPa.

## A2   Sponge layer Rayleigh damping

Following HS94 and W98, we use the inverse timescales $k_v(p)$ and $k_T(\phi, p)$ for Rayleigh damping of the velocity $\boldsymbol{v}$ and
relaxation temperature $T$ toward $T_\text{eq}$, respectively. In addition, we also add a second Rayleigh damping mechanism in the "sponge layer" (so-called since it acts to absorb vertically propagating waves near the model top). The vertical profile that we choose for the damping strength follows the implementation of Harris et al. (2021) (their Eq. (8.15)), having a monotonic onset from ~100 Pa to the model top:

$$k_s(p) = k_0 \sin\left(\frac{\pi}{2} \frac{\log(\eta_c/\eta)}{\log(\eta_c/\eta_T)}\right)^2. \tag{A2}$$

Here, the normalized pressure coordinate is $\eta \equiv p/p_0$, with $p_0 = 1000$ hPa. We define an onset position at $\eta_c = (1\,\text{hPa})/p_0$, and the normalized pressure at the model top as $\eta_T \equiv p_\text{top}/p_0$. The maximum strength of the damping is set via $k_0 = 1/(3\,\text{days})$.

Given these modifications, the wind and temperature tendencies will be updated at each physics timestep by

$$\frac{\partial T}{\partial t} = -k_T(\phi, p)(T - T_\text{eq}) \tag{A3}$$

$$\frac{\partial \boldsymbol{v}}{\partial t} = (-k_s(p) - k_v(p))\,\boldsymbol{v} \tag{A4}$$

which is the totality of parameterized forcings in our model (in absence of volcanic injections). The vertical profile of the total wind damping $(k_s(p) + k_v(p))$ is shown in Fig. 1, panel (a).

## Appendix B:  Recommendations for model parameter tuning

This section provides recommendations for re-tuning the idealized volcanic forcing model presented in this work. This information may be useful if the forcing set is to be run on a new dynamical core resolution, or if the model parameters are altered
to represent a different aerosol injection scenario. The results shown in Sect. 4.2 only represent the Pinatubo-like parameter configuration given in Table 1 for a grid of nominal 200 km horizontal spacing (the ne16pg2 grid).

In our original implementation, the parameter tuning was done by manually perturbing parameter values after a preliminary estimate, running simulations, and observing the result of certain target metrics. In practice, the tuning process is an iterative



one, since changing the extinction coefficients changes the local aerosol heating rates, which in turn changes the local circu-
lation and thus plume transport. Achieving a plume that leads to both a realistic stratospheric distribution and global mean
temperature anomalies was the goal.

Each subsection below describes the tuning of a certain effect, first listing the relevant parameters, the target metric used,
and the method of the initial estimate. We suggest that future tuning efforts of our parameterizations follow these procedures.
In what follows, we refer to a "passive" run as one where the tracer injection and sulfate production occurs as described in
Sect. 3.1-3.2, but all radiative feedback is disabled, and the tracers are only transported.

### B1 SW mass extinction coefficients

| | |
|---|---|
| **Parameters:** | $b_{SW,ash}$, $b_{SW,SO2}$, $b_{SW,sulfate}$ |
| **Target metric:** | maximum zonal-mean AOD |
| **Initial estimate method:** | passive simulation runs |

In tuning the shortwave extinction coefficients, we can make a preliminary constraint of $b_{SW}$, such that the resulting AOD $\tau_i$
is representative of post-Pinatubo observations. Zonal-mean AODs observed in the months and years following the Pinatubo
eruption peaked near 0.2-0.5 (Dutton and Christy, 1992; Stenchikov et al., 1998; Mills et al., 2016), and passive runs of our
injection protocol yield maximum zonal-mean column mass burdens (as a sum of all species) which peak near $2 \times 10^7$ kg
approximately three weeks post-injection near the equator. Constraining columns of this mass burden to have an AOD of 0.35,

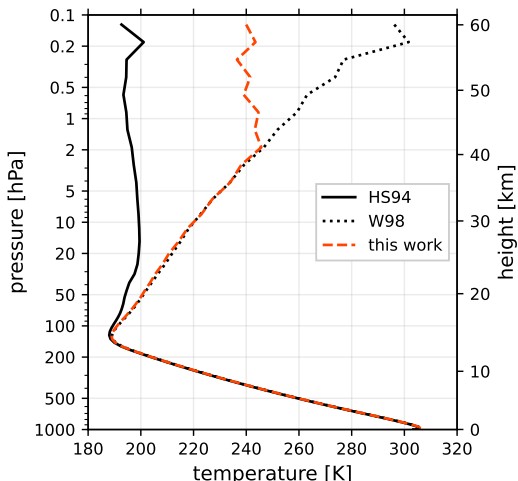

**Figure A1.** Zonal-mean tropical temperature profiles resulting from E3SMv2 runs of the HS94 (solid black), W98 (dotted black), and our
modified HSW (dashed red) forcing schemes, averaged over $-7°$S to $7°$N. The runs were done at the ne16pg2 resolution for 10 years, with
the first five years discarded as spin-up, and the time mean of the latter five-year period shown here. The height axis is derived from the
geopotential height of the modified HSW run, with the same averaging performed.





Eq. (23) then suggests

$$\tau_i = 0.35 = b_{\text{SW}} \frac{M_i}{a_i}$$

$$\implies b_{\text{SW}} = \frac{0.35 a_i}{2 \times 10^7 \, \text{kg}} = 700 \, \frac{\text{m}^2}{\text{kg}} \tag{B1}$$

where we took $a_i = (200 \times 200) \, \text{km}^2$, consistent with a ~2 degree resolution near the equator. Starting with this initial estimate, we then manually and iteratively altered the $b_{\text{SW}}$ parameters until converging upon the desired peak zonal-mean AOD (see Fig. 8), with the final parameter values given in Table 1. These final values are an acceptable starting estimate for new tuning efforts. Since Eq. (B1) involves a factor of (area)/(mass), the $b_{\text{SW}}$ parameters should be, in principle, independent of resolution.

However, higher resolutions could resolve finer, higher-density structures in the tracer fields and might therefore be indirectly dependent on the grid spacing.

## B2   Surface heat transfer efficiency

| | |
|---|---|
| **Parameters:** | $\zeta$ |
| **Target metric:** | minimum mean surface temperature anomaly |
| **Initial estimate method:** | derived constraint from literature |

As we do not a priori know the relationship between surface cooling rates and the realized surface temperature anomaly

in the HSW atmosphere, we make a preliminary tuning of $\zeta$, assuming a known maximum negative cooling rate. In their model experiments of the Pinatubo initial plume dispersion, Stenchikov et al. (2021) observed that spatial-mean values of the surface cooling in the equatorial belt from $0°$-$15°$N post-injection are around $\Delta T = -0.02$ K/day (their Fig. 6), which is also qualitatively consistent with Stenchikov et al. (1998) and Ramachandran et al. (2000). In this region, we have already roughly constrained $\tau$ to be near 0.35 in Eq. (B1). Solving Eq. (35) for $\zeta$ in this case gives:

$$\left( -0.02 \, \frac{\text{K}}{\text{day}} \right) = \zeta \frac{1}{c_p} \frac{a_i}{m_{i,\text{surf}}} \frac{(1 \, \text{day})}{(86400 \, \text{s})} I_{\text{SW}} \left( \exp(-0.35) - 1 \right)$$

$$\implies \zeta^{-1} = \frac{1}{c_p} \frac{a_i}{m_{i,\text{surf}}} \frac{(1 \, \text{day})}{(86400 \, \text{s})} \frac{1}{\left( -0.02 \, \frac{\text{K}}{\text{day}} \right)} I_{\text{SW}} \left( \exp(-0.35) - 1 \right) . \tag{B2}$$

Using $I_{\text{SW}}$ from Eq. (5) at latitude $\phi = 0$, and $m_{i,\text{surf}} = (\Delta p_{i,k} a_i / g)$ with $\Delta p_{i,k} = 10$ hPa (corresponding to ~100 meters or $\kappa = 2$ for Eq. (37) in E3SMv2), this gives

$$\zeta \approx 1.5 \times 10^{-3}$$

which is independent of horizontal resolution, since $a_i$ cancels in Eq. (B2) (though it is dependent on vertical resolution). The tuned value of $4.0 \times 10^{-3}$ is several times larger than this estimate, since we ultimately required average tropical cooling rates significantly lower than $-0.02$ K day$^{-1}$ (see Fig. 8, panel (a)) to obtain a significant surface-level temperature anomaly near $-1$ K. This requirement may simply be due to the strength of the HSW temperature relaxation at the lowest model levels needing a stronger forcing to overcome.



### B3 LW mass extinction coefficients

| | |
|---|---|
| **Parameters:** | $b_{\mathrm{LW,ash}}, b_{\mathrm{LW,SO2}}, b_{\mathrm{LW,sulfate}}$ |
| **Target metric:** | maximum mean stratospheric temperature anomaly, and lofted sulfate plume height |
| **Initial estimate method:** | passive simulation run, and derived constraint from literature |

In making initial estimate of the LW mass extinction parameters, we first simplify Eq. (28) to an approximate form where the aerosol radiative shadowing effect is not allowed (assuming $I_{\mathrm{LW}}$ is incident on all vertical positions of the column):

$$\Delta I_{i,k} = I_{\mathrm{LW}} \left[ 1 - \exp\left( -b_{\mathrm{LW}} \frac{q_{i,k}\,\Delta p_{i,k}}{g} \right) \right] . \tag{B3}$$

This simplified attenuation implies a stratospheric heating rate in the form of Eq. (30) of

$$\begin{aligned} \Delta T_{i,k} &= \frac{1}{c_p} \frac{a_i}{m_{i,k}} I_{\mathrm{LW}} \left[ 1 - \exp\left( -b_{\mathrm{LW}} \frac{q_{i,k}\,\Delta p_{i,k}}{g} \right) \right] \\ &\approx -\frac{1}{c_p} \frac{a_i}{m_{i,k}} I_{\mathrm{LW}}\, b_{\mathrm{LW}} \frac{q_{i,k}\,\Delta p_{i,k}}{g} , \end{aligned} \tag{B4}$$

where the approximation $e^x \approx (1+x)$ was used. The estimate of $b_{\mathrm{LW}}$ can now proceed analogously to Sect. B1-B2, by inverting Eq. (B4) for $b_{\mathrm{LW}}$. From observations by previous modeling studies (Stenchikov et al., 2021; Ramachandran et al., 2000; Stenchikov et al., 1998), we expect monthly-mean zonal-mean values for the stratospheric heating rate during month three post-injection to approach $\Delta T = 0.3$ K day$^{-1}$ in the tropics. Passive runs of our injection protocol yield sulfate monthly-mean zonal-mean mixing ratios at this time and location of about $10^{-6}$ kg kg$^{-1}$. The preliminary $b_{\mathrm{LW}}$ estimate is then

$$\implies b_{\mathrm{LW}} = \frac{\left( 0.3 \frac{\mathrm{K}}{\mathrm{day}} \right)}{\left( 10^{-6}\,\frac{\mathrm{kg}}{\mathrm{kg}} \right)} \frac{g\, c_p\, m_{i,k}}{a_i\, I_{\mathrm{LW}}\, \Delta p_{i,k}} \frac{(1\text{ day})}{(86400\text{ s})}\, \frac{\mathrm{m}^2}{\mathrm{kg}} . \tag{B5}$$

Evaluating this form with $I_{\mathrm{LW}}(\phi = 0)$ from Eq. (4), and using $m_{i,\mathrm{surf}} = (\Delta p_{i,k} a_i / g)$ gives

$$b_{\mathrm{LW}} \approx 6.2\, \frac{\mathrm{m}^2}{\mathrm{kg}} . \tag{B6}$$

Comparing this estimate with Eq. (B1), our formulation implies that the aerosols are much more efficient at attenuating shortwave radiation (by scattering) than longwave radiation (by absorption).

From this estimate, we then manually and iteratively adjust the three $b_{\mathrm{LW}}$ parameters for ash, SO$_2$, and sulfate. As suggested in Sect. 3.5, this tuning is done with two target metrics in mind: (1) $b_{\mathrm{LW,\,sulfate}}$ is tuned to give rise to maximum stratospheric temperature anomalies of 2-3 K, and (2) $b_{\mathrm{LW,\,ash}}$ is tuned to control the initial lofting of the fresh, dense plume, such that the aged sulfate population converges upon the 20-30 km vertical layer. The parameter $b_{\mathrm{LW,\,SO2}}$ contributes to both the initial lofting and short-term temperature anomalies. Thus, the final tuned parameters (given in Table 1) arrive at very different values.

The tuning process would be easier, and a higher initial injection height of 18-20 km could be supported, if the degeneracy between these three extinction parameters were removed. We recommend having the SO$_2$ tracer instead behave as a radiatively passive tracer, acting only as the vehicle for sulfate production. In this case, the LW mass extinction coefficients for ash and





sulfate would truly be independent knobs for the lofting height, and long-term temperature anomalies, respectively. We would consider this tuning choice an improvement of the parameterization.

*Author contributions.* JPH wrote the Fortran subroutines that implement these parameterizations in E3SMv2, conducted the ensemble model experiments, tuned the model parameters, and wrote the manuscript. CJ originally suggested the implementation of volcanic forcing in an
idealized framework, and collaborated on implementation and analysis. HB collaborated on the LW and SW radiative forcing formulation, and reviewed early manuscript drafts. BRH enabled the FIDEAL (including HSW) forcing set in E3SMv2, and provided support on integrating our parameterizations with the E3SMv2 physics infrastructure. DLB guided the model design, consulted on target atmospheric impact metrics, and collaborated on devising the simulation ensemble and output strategies. JLH motivated the development of the HV and LV ensemble generation strategies, as well as the forms for the tracer sources and sinks. HB and DLB contributed text to the introduction and
conclusion. All co-authors reviewed the manuscript.

*Competing interests.* The authors declare that they have no conflict of interest

*Acknowledgements.* The work was supported by the Laboratory Directed Research and Development program at Sandia National Laboratories (SNL), a multimission laboratory managed and operated by National Technology & Engineering Solutions of Sandia, LLC, a wholly owned subsidiary of Honeywell International Inc., for the U.S. Department of Energy's National Nuclear Security Administration under
contract DE-NA0003525. This written work is co-authored by employees of NTESS. The employees, not NTESS, owns the right, title and interest in and to the written work and is responsible for its contents. Any subjective views or opinions that might be expressed in the written work do not necessarily represent the views of the U.S. Government. The publisher acknowledges that the U.S. Government retains a non-exclusive, paid-up, irrevocable, world-wide license to publish or reproduce the published form of this written work or allow others to do so, for U.S. Government purposes. The DOE will provide public access to results of federally sponsored research in accordance with the
DOE Public Access Plan. The University of Michigan (UM) researchers were supported by an SNL subcontract, award number 2305233. This research used resources of the National Energy Research Scientific Computing Center (NERSC), a U.S. Department of Energy Office of Science User Facility located at Lawrence Berkeley National Laboratory, operated under Contract No. DE-AC02-05CH11231 using NERSC award BER-ERCAP0022865. The team also greatly benefited from the discussions with the UM scientist Owen Hughes and the CLDERA SNL project members Jerry Watkins, Andrew Steyer, Benjamin Wagman, Irina Tezaur, Kara Peterson, Laura Swiler, and the
greater CLDERA team. The analyses presented in the Results section of this manuscript made extensive use of the NumPy (Harris et al., 2020), Cartopy (Elson et al., 2023), MetPy (May et al., 2022), and xarray (Hoyer and Hamman, 2017) Python packages.



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
