# Peer review of "Localized injections of interactive volcanic aerosols and their climate impacts in a simple general circulation model"

_EGUsphere, 2024_

## Author Comment (AC1)

**Authors' Response to Reviews of**

**HSW-V v1.0: localized injections of interactive volcanic aerosols and their climate impacts in a simple general circulation model**

Joseph Hollowed, Christiane Jablonowski, Hunter Y. Brown, Benjamin R. Hillman, Diana L. Bull, and Joseph L. Hart

*egusphere-2024-335*
* * *
RC: *Reviewer Comment*,     AR: Author Response,     ☐ Manuscript Text

**1. Reviewer #2**

**1.1. Author Comments**

We thank the reviewer for carefully reading our manuscript and providing very constructive suggestions for improvement. Each comment below appears as a reviewer comment (RC) followed by an author response (AR). Closed boxes show text from the manuscript. Red text with strikethrough represents deleted text, and blue text with wavy underlining represents new text. Section numbers refer to those as they appear in the updated manuscript (for example, some Appendix section numbers have changed).

The biggest changes made are in our response to Comment 1, where we added new paragraphs of text, a new appendix section, and a new figure in support of a discussion of the Brewer-Dobson circulation in our model. In our responses to Comment 2 and Comment 11, we have added several new references. The remaining comment responses consist of small edits and clarifications.

**1.2. Comment 1**

RC: *My only more general comment relates to the lack of any mention of the Brewer-Dobson circulation (BDC), which is the main process which controls the transport and spread of stratospheric aerosol. Around line 496, some discussion of the tropospheric large scale circulation is present, which may be relevant to the simulations given the rather low injection height used in the "Pinatubo-like" simulations, but the general utility of this model set-up will depend in part on the fidelity of the BDC: in terms of general features like isolation of the tropical pipe, large-scale mixing in the extratropical stratosphere, and polar subsidence. If an assessment of the stratospheric meridional circulation in the HSW model configuration is given in other studies, it would be useful to summarize some of that work in the introduction. If not, maybe the authors can provide some guidance based on their own results.*

AR: We strongly agree that the fidelity of the BDC is relevant to the utility of this model. While we did attempt to introduce some discussion on the global circulation in Section 4.2, as the reviewer has noted, we agree that the focus on the Hadley cell is mostly relevant to the troposphere, and that a broader discussion on the residual circulation in the stratosphere is missing. At least one prior study has shown transformed Eulerian mean (TEM) analysis results for a HSW atmosphere (Yao & Jablonowski, 2016), but we are not aware of a study that shows the residual circulation in particular, and of course none which include our modified HSW implementation near the model top.

To this end, we have performed an analysis to obtain the residual velocity components and streamfunction in

the stratosphere, averaged over five years of an HSW run with no volcanic injection present. A new appendix containing a new figure (Appendix B, Figure B1) show the results. Section 4.2 also now includes an extended discussion on the global circulation which references this appendix. Specifically, we show that the global stratospheric circulation is qualitatively consistent with the equinoctial (hemispherically symmetric) state of the BDC in nature. This of course differs from the solstice condition of the BDC which was manifest during the historical Pinatubo event in June of 1991. This difference is important to understand, but does not diminish the utility of our model, in our opinion. Still, we have included a sentence which suggests that the HSW relaxation temperature could, in principle, be replaced with an asymmetric form if desired, which has precedent in prior studies.

The new paragraph in Section 4.2 is as follows:

[revised manuscript text omitted]

**1.3. Comment 2**

**RC:** *Line 29: there are of course many different stratospheric aerosol models used by groups around the world. Therefore it could be here widen the scope of references on such model beyond one single model. A possibility might be to include a reference to a study which includes multi-model comparison (e.g., Clyne et al., 2021) and/or a model-focused review paper (e.g., Timmreck, 2012).*

AR: We have added a citation to Clyne et al., 2021 alongside the existing citation to Zanchettin et al., 2016, which seemed appropriate. We have also reworded the final sentence of this paragraph and added a citation to the Timmreck, 2012 paper. We thank the reviewer for bringing these works to our attention.

> Prescribed forcing approaches might be chosen for their computational affordability, though they are also used to facilitate climate model intercomparisons by standardizing the forcing scheme (Zanchettin et al., 2016; Clyne et al., 2021).
>
> ...
>
> A review of these Reviews of the wide array of modeling choices for volcanic forcings is presented in made by different ESMs are presented in Timmreck (2012) and Marshall et al. (2022).

**1.4. Comment 3**

**RC:** *Line 150: Reference here to the HSW "forcing set" seems like a different definition of "forcing" as the term is applied to volcanic aerosol forcing.*

**AR:** We appreciate the comment from the reviewer and have considered alternative language here. Ultimately, we feel that this usage of "forcing" is correct and consistent with the literature. It may be true that these are different kinds of forcings from a physical perspective; the volcanic aerosol forcing refers to a real physical process being modeled, while the "HSW forcing" refers to an "artificial" process of relaxation toward the reference temperature profile. Still, they are treated identically on paper (mathematically) and in the model, as they both refer to an additive term to the temperature tendency at each model timestep.

For what it's worth, we have replaced a few occurrences of the term "forcing set" in a few sections, since this sounds more specific but was used inconsistently. Beyond this, we did not replace the general usage of "HSW forcing" throughout the rest of the manuscript. The changes were as follows:

- Line 91: replaced "forcing set" with "scheme".
- Line 107: replaced "forcing set" with "set of forcing functions".
- Line 152: "HSW frocing set" was replaced with "HSW atmosphere".
- Line 628: "forcing set" was replaced with "parameterizations".
- Line 708: "forcing set" replaced with "configuration".

**1.5. Comment 4**

**RC:** *Line 222: "timestep n"?*

**AR:** The word "time" was replaced with "timestep" in this sentence.

**1.6. Comment 5**

**RC:** *Line 289: I guess it should be "either absorption or scattering" or "absorption and scattering", the latter being more generally accurate.*

**AR:** We agree. The sentence was changed as follows:

> A single aerosol species can contributes to extinction of transmitted radiation by either absorption and scattering,...

**1.7. Comment 6**

**RC:** *Line 300: Probably useful to specify you refer specifically to the SW AOD here. There is also a LW AOD even if it isn't directly considered.*

**AR:** We decided that this sentence is unnecessary, and instead added a new sentence to the end of this section which specifies that all further references of the AOD refer specifically to the shortwave AOD. The changes are as follows:

> $b_{\mathrm{LW}}$ will be used for the extinction of longwave (LW) radiation, which is assumed to be entirely absorption, and $b_{\mathrm{SW}}$ will be used for the extinction of shortwave (SW) radiation, which is assumed to be entirely scattering:
>
> $$b_{\mathrm{LW}} \equiv (b_e \text{ for the longwave band}),$$
> $$b_{\mathrm{SW}} \equiv (b_e \text{ for the shortwave band}).$$
>
>
>
> ...
>
> For a column with a model top at $z_{\mathrm{top}}$, the dimensionless SW AOD $\tau$ at a height $z$ is obtained by
>
> ...
>
> We also define a shorthand for the cumulative SW AOD at the surface as $\tau_i \equiv \tau(z=0)$. After summing over $k$ for this case, we have the usual result (Petty, 2006) that each remaining term is just the total column mass burden $M_i$ of the tracer, scaled by the mass extinction coefficient $b_{\mathrm{SW}}$ and column area $a_i$,
>
> $$\tau_i = \sum_k b_{\mathrm{SW}} \frac{q_{i,k}\Delta p_{i,k}}{g} = \sum_k b_{\mathrm{SW}} \frac{q_{i,k} m_{atm,i,k}}{a_i} = b_{\mathrm{SW}} \frac{M_i}{a_i}. \tag{23}$$
>
> Hereafter, "AOD" will refer specifically to the column-integrated SW AOD defined in Eq. (23).

**1.8. Comment 7**

**RC:** *Line 328: It would be good to be explicit about ignoring heating from near-IR solar radiation which was shown by Stenchikov et al. (1998) to be a contributing factor to the total aerosol heating.*

**AR:** We thank the reviewer for raising this issue; we did not appreciate the difference between near-IR and LW heating, and/or did not describe carefully enough the tuning process. In fact we do tune our heating rates (by way of the LW mass extinction coefficients) to those presented in Stenchikov et. al. (1998) as "total" heating rates (their Figure 10, bottom row), as is described in Appendix C3. In other words, our tuning is implicitly accounting for the near-IR contribution, though we do not name it as such. We have added an explicit mention of this fact in Appendix C3 as follows:

> From observations by previous modeling studies (Stenchikov et al., 2021; Ramachandran et al., 2000; Stenchikov et al., 1998), we expect monthly-mean zonal-mean values for the stratospheric heating rate during month three post-injection to approach $\Delta T = 0.3$ K day$^{-1}$ in the tropics. Note that in the

works cited for this figure, this is the *total* heating rate due to contributions of visible, near-infrared, and infrared radiation. In this work, though we refer to this heating effect specifically as "longwave", we are tuning to the *total* heating rate of 0.3 K day$^{-1}$.

As well as in Section 3.5 as follows:

The longwave attenuation mechanism of the model is tuned to produce realistic stratospheric heating rates by sulfate aerosols. The mass extinction coefficient $b_{\mathrm{LW}}$ for sulfate is instrumental in tuning the long-term mean temperature anomalies. We note that while we refer to this heating mechanism specifically as a "longwave attenuation", the tuning process implicitly accounts for heating contributions from the near-infrared radiation as well (see Appendix C3)...

**1.9. Comment 8**

**RC:** *Line 446: missing "we" in sentence*

**AR:** This sentence was reworded and moved to the end of the section as follows:

In Sect.4.2, only show results from an HV ensemble, though we encourage future experiments to present their ensemble generation methods in these terms.

...

For the purposes of the present work, the model results of Sect. 4.2 are shown only for a HV ensemble. We encourage future studies using this model to present their ensemble generation methods in these terms.

**1.10. Comment 9**

**RC:** *Fig. 7 caption describes time axis as "time since eruption" which appears inaccurate.*

**AR:** This sentence in the caption was reworded as follows:

Time values are shown as days since eruptionThe eruption occurs at day 180.

**1.11. Comment 10**

**RC:** *Fig. 8 caption: the black marker should be said to denote the time and latitude of injection.*

**AR:** This was corrected by replacing "vertical center" with "latitude".

**1.12. Comment 11**

**RC:** *Line 558: here and/or in introduction, it would be useful to make reference to the work of DallaSanta et al., (2019) who used a model hierarchy to investigate the dynamical impacts of volcanic forcing, using a much simpler prescribed aerosol forcing.*

**AR:** We thank the reviewer for bringing this paper to our attention. It will be a very valuable citation and reference for this current manuscript as well as future work. We have added a reference to DallaSanta et al. (2019) in both Section 1 as follows:

> Simpler techniques prescribe radiative aerosol properties directly from an external dataset or analytic forms (e.g., see DallaSanta et al. (2019); Toohey et al. (2016); Eyring et al. (2013); Gao et al.25 (2008); Kovilakam et al. (2020)).

and in Section 5 as follows:

> This work is a new addition to an idealized AGCM model hierarchy that can be used to study phenomena in isolation. Examples of this model hierarchy include Sheshadri et al. (2015) and Hughes and Jablonowski (2023) who studied the effects of topography on the atmospheric flow, or Polvani and Kushner (2002) and Gerber and Polvani (2009) who assessed polar jets and hemispheric asymmetry. Other idealized configurations focus on simple moist flows with moisture feedbacks (Frierson et al., 2006; Thatcher and Jablonowski, 2016), tropical cyclones (Reed and Jablonowski, 2012), tracer-based cloud micro-physics (Frazer and Ming, 2022; Ming and Held, 2018), age-of-air tracers (Gupta et al., 2020), the Madden-Julian oscillation (MacDonald and Ming, 2022), or climate-change forcing (Butler et al., 2010). In addition, this work builds upon previous idealizations of volcanism using simpler prescribed forcing approaches. This includes Toohey et al. (2016) who provide a set of zonally-symmetric volcanic aerosol optical properties tuned to observational data, and DallaSanta et al. (2019) who subjected a set of atmospheric models of increasing complexity to a prescribed aerosol forcing in the form of controlled solar dimming, and steady, zonally uniform lower-stratospheric temperature tendencies.

**1.13. Comment 12**

**RC:** *Line 566: A stratovolcano is a particular type of volcano, not one that produces injections of sulfur to the stratosphere.*

**AR:** This was corrected by replacing "stratovolcano eruptions" with "volcanic eruptions".

**1.14. Comment 12**

**RC:** *Line 588: a little copy editing required*

**AR:** This sentence was edited as follows:

> The three implementations exhibit no difference in tropopause structure. Above the tropopause, the HS model approaches a constant temperature near 200 K, while W98 and the modified HSW forcings share a lapse rate of approximately 2.6 K km$^{-1}$ until  2 hPa , where they diverge.

We also moved this paragraph to the end of Appendix A1, rather than the end of Appendix A, where it was out-of-place.

---

## Author Comment (AC2)

**Authors' Response to Reviews of**

**HSW-V v1.0: localized injections of interactive volcanic aerosols and their climate impacts in a simple general circulation model**

Joseph Hollowed, Christiane Jablonowski, Hunter Y. Brown, Benjamin R. Hillman, Diana L. Bull, and Joseph L. Hart
*egusphere-2024-335*
* * *
**RC:** *Reviewer Comment*,      AR: Author Response,      ☐ Manuscript Text

**1.  Reviewer #1**

**1.1.  Author Comments**

We thank the reviewer for the careful reading of our manuscript and the useful feedback. Each comment below appears as a reviewer comment (RC) followed by an author response (AR). Closed boxes show text from the manuscript. Red text with strikethrough represents deleted text, and blue text with wavy underlining represents new text. Section numbers refer to those as they appear in the updated manuscript (for example, some Appendix section numbers have changed).

Our responses to Comment 1, 2 and 3 consist of important text edits to make more clear the relationship of this work to the broader field of climate-attribution science. Our response to Comment 4 identifies a mistake that was present in the manuscript.

**1.2.  Comment 1**

**RC:** *The authors do not show that this first-order treatment of transport, temperatures, radiation, and aerosol processes create a trustworthy climate-attribution environment. One criticism, for example, might stem from a lack of a quasi-biennial oscillation in the model. This deficiency both eliminates one way in which volcanic eruptions impact circulation and one mode of dynamical variability that impacts the transport of volcanic aerosols and their precursors.*

**AR:** We appreciate this feedback from the reviewer, and have thought carefully about how to better express the application of our model to the climate-attribution problem. We would like to emphasize that this model configuration is intended to be a trustworthy environment in which to *develop* new climate attribution methodologies, not one in which the specific Pinatubo impacts are accurately modeled. Pathways of impact will exist even if the general circulation does not specifically represent that of the historical Pinatubo scenario, or even that of the Earth's observed climatology. This is an intentional simplification. To be clear, the idealized environment is being employed to eliminate some of the complexity of a fully coupled / full physics Earth System Model (ESM) whilst still preserving the progressive multi-variate pathway through which impacts arise, such that the implementation of novel methodologies can be verified as operating correctly. This lowers the risk of moving to more complex problems in which realistic representations are present.

We have made some changes to the language in Section 1 in the hopes that this nuance is more clear. Specifically, we replaced occurrences of the phrase "attribution problem", since it may have implied something specific and different from what we intended:

…

Prescribed and prognostic methods have also been applied to model other forms of sulfur-based radiative forcing, with significant research recently being devoted to stratospheric aerosol injection (SAI) climate-change intervention activities (Crutzen, 2006; Tilmes et al., 2018, 2017; McCusker et al., 2012).  . One key goal of SAI research is to quantify the causal connections between an observed climate impact, and an upstream forcing source, i.e. to attribute the SAI source as the cause of a detected, anomalous atmospheric response. Volcanoes are a natural analog to SAI, and thus offer  an avenue for developing novel attribution methods of quantifying these causal connections.

The climate impacts that are most  societally-relevant tend to be spatially localized (e.g. droughts, heat waves, or fires) and located downstream from their associated sources (e.g. volcanoes, or other solar radiation modification)  by multiple causal connections. "Multi-step attribution" involves a sequence of single-step attribution analyses, but is generally not employed, as the single weakest attribution step limits its confidence (Hegerl et al., 2010). Therefore, there is a need for  novel multi-step attribution techniques in both climate change studies (Burger et al., 2020) and climate intervention studies (National Academies of Sciences, 2021; Office of Science and Technology Policy (OSTP), 2023)  that overcome these issues to enable attribution of societally-relevant impacts.

…

Accordingly, we suggest that a  new idealized representation of prognostic volcanic forcing within a highly simplified atmospheric environment would be a useful testbed for the development of novel multi-step attribution methods (i.e. constructing relationships between stratospheric aerosol forcing and atmospheric temperature perturbations).

…

In addition, see our response to Comment 2 for a few more changes relevant to this comment.

As to the issue of the QBO specifically, we hope that these clarifications demonstrate that an accurate representation of the QBO (and other specific modes of climate variability) are not necessarily required in order for the model to serve as an climate-attribution development testbed. We agree with the reviewer that the lack of a QBO in our model will change the aerosol transport, and thus the specific atmospheric impacts of the volcanic forcing, with respect to the historical event. However, we do not think that this fact necessarily challenges the utility of our model as we have presented it.

Having said this, we do think that it is worthwhile to emphasize more explicitly the lack of a QBO in our configuration, and what implications this has on the downstream impact development. We would also like to emphasize to readers that it would be possible to activate an auxiliary parameterization which nudges the equatorial winds toward a realistic QBO, if desired. To this end, we added a new paragraph at the end of Section 2.2:

> In the tropical stratosphere, easterlies with speeds up to $-30$ m s$^{-1}$ dominate. Note that while the tropical stratospheric winds will vary about this average, the HSW atmosphere does not include any kind of regular quasi-biennial oscillation (QBO) analog. Yao and Jablonowski (2016) showed that whether or not a QBO spontaneously develops in an HSW configuration will largely depend on the dynamical core in use. For a spectral element (SE) dynamical core, they observed that wave forcing was never strong enough to cause a reversal of the tropical stratospheric winds. The same conclusion appears to hold for our configuration of E3SMv2. Despite this, the QBO may be a desirable target for future studies employing this model configuration, as it has been shown that the QBO phase is a significant modulator of the volcanic climate response (Thomas et al., 2009). We do not consider this issue further in the present work, but note that it could be possible to prescribe a QBO by nudging the horizontal winds toward a specified reference state (as has been done for e.g. the Whole Atmosphere Community Climate Model (WACCM) by Matthes et al. (2010)).

**1.3. Comment 2**

**RC:** *A related issue stems from the assertion that, on line 62, "the goal is not to accurately replicate any particular historical eruption". This assertion seems at odds with the rest of the paper which is dominated by an example of tuning parameters in order replicate the specific (and unique) eruption of Pinatubo in 1991. This highly idealized setup requires tuning these parameters; the authors should clarify how this scheme could be used in a general fashion that isn't based on tuning parameters to a specific eruption. One suggestion for future work might be to tune the parameters to some kind of average of many eruptions.*

AR: This is an issue that we may not have been clear enough about in the manuscript, so we appreciate the reviewer bringing it to our attention. When we say that "the goal is not to accurately replicate any particular historical eruption", what we really mean is that our model is not attempting to capture the specific *observed atmospheric response* to the Pinatubo eruption. This is necessarily true, since our atmosphere is hemispherically symmetric, and does not represent certain atmospheric modes that were present during the historical event (e.g. the QBO). Despite this, we still chose to tune the *volcanic forcing itself* toward a specific exemplar. In other words, the goal was to represent plausible atmospheric impacts of a Pinatubo-like event, and not to be predictive of the observed impacts themselves.

We have adjusted the text in Section 1 to be more clear about this intention:

> Our approach sacrifices realism by design. The goal is not to simulate an accurate post-eruption climate of a particular historical volcanic event, but rather to produce a plausible realization of a generic volcanic eruption, simulated with a minimal forcing set.
>
> ...
>
> Our model isolates a single volcanic event from any other external source of forcing or variability, and allows the flexibility to be embedded in a simplified atmospheric environment.  Though the implementation is generic, we present here a particular tuning of the parameterizations for an eruption similar in character to the 1991 eruption of Mt. Pinatubo, and the subsequently observed impacts...

We note that the final paragraph in Section 5 does describe more general usage of our parameterizations, as the reviewer has suggested, though we have adjusted the text slightly to be more explicit:

> We illustrated that our implementation can be used to mimic the spatio-temporal temperature anomaly signatures of large volcanic eruptions, and presented one specific parameter tuning that gives rise to a Pinatubo-like event... Nevertheless, the formulation remains flexible to modifications. Our parameterizations can be tuned toward eruption scenarios other than the 1991 Mt. Pinatubo event. They can also support any number of co-injected tracer species, concurrence of multiple eruptions, and injections at any latitude and height. In fact, the description is generic enough that by replacing the vertical and/or temporal injection profiles, we could imagine simulating the aerosol direct-effect of various localized emission events of the troposphere (e.g. wildfire smoke) or the stratosphere (e.g. geoengineering SAI experiments) in an idealized model configuration.

**1.4. Comment 3**

**RC:** *Specific to the Pinatubo eruption, the authors should expand on parameter choices. Observations (especially early on) of Pinatubo are uncertain. That being said, a plume center of mass at 14km for Pinatubo is extremely low. It is implied that this is due to unrealistic plume rise observed in this system—could the authors expand on that?*

AR: We agree that this figure is a bit jarring to familiar readers, and that more discussion is warranted. This was an outcome of the tuning process, and was required to obtain the desired long-term temperature anomalies. As we state in Section 3.5:

> The longwave attenuation mechanism of the model is tuned to produce realistic stratospheric heating rates by sulfate aerosols. The mass extinction coefficient $b_{LW}$ for sulfate is instrumental in tuning the long-term mean temperature anomalies ... Not as obvious is the importance of $b_{LW}$ for the very short-lived ash tracer. The lofting speed of the plume will be controlled by the aggressive early heating of ash in the fresh plume (Stenchikov et al., 2021), since the initial ash mass loading (50 Tg) is dominant over that of $SO_2$ (17 Tg). As such, the mass extinction coefficient for ash serves as the main tuning parameter which controls the settling height of the aged aerosols.

In fact, the final sentence in this quote is not quite correct, and SO2 still contributes significantly to the initial plume heating, and subsequent lofting. When we tuned the mass extinction coefficient for ash, in order to achieve a realistic settling height near 25 km, we did *not* also tune the SO2 mass extinction coefficient. In hindsight, this could have been done differently. This essentially means that we heat the initial plume more than we really intend to, which we must accommodate for by lowering the initial injection height. To avoid this, we could have tuned the SO2 $b_{LW}$ simultaneously with that of ash, or could have simply set $b_{LW,SO2} = 0$, and thus controlled the heating of the fresh plume by ash alone. We attempted to explain this in Appendix C3, and specifically recommend changing this parameter choice in future usage of the model, if a higher injection height is desired:

> The tuning process would be easier, and a higher initial injection height of 18-20 km could be supported, if the degeneracy between these three extinction parameters were removed. We recommend having the SO2 tracer instead behave as a radiatively passive tracer, acting only as the vehicle for sulfate production. In this case, the LW mass extinction coefficients for ash and sulfate would truly be independent knobs for the lofting height, and long-term temperature anomalies, respectively. We would consider this tuning choice an improvement of the parameterization.

We did not find this issue to be problematic enough to warrant re-tuning the model. This is because the signature of the forcing and associated atmospheric impacts of the mixed (zonally symmetric) aerosol distribution would not change, which was our priority. In addition, we do not think that the current configuration would preclude an analysis which focuses more on the initial plume evolution, as the modeled scenario is still physically plausible, even if not perfectly reminiscent of the Pinatubo event (also see responses to Comment 1 and Comment 2).

We have ensured that all of this is more clear to the reader. In particular, we added a few more words about this issue to the main text, rather than only appearing in the appendix, where it might be missed. First, the text in Section 3.5 has been adjusted:

> ...
>
> Not as obvious is the importance of $b_{LW}$ for the very short-lived ash tracer. Though radiative forcing by ash does not directly contribute to the eventual stratospheric temperature anomalies, it does control the mechanism by which the aerosols are delivered to the lower stratosphere (Stenchikov et al., 2021). The lofting speed of the dense, fresh plume will be controlled by the aggressive  heating of ash, which is the dominant component of the initial injection. As such, the mass extinction coefficient for ash serves as the main tuning parameter which controls the settling height of the aged aerosols. Meanwhile, $SO_2$  participates both in the initial lofting of the plume, as well as the short-term temperature anomalies for the first couple months. This behavior by $SO_2$ creates some degeneracy in the longwave extinction tuning parameters which could be avoided with a slight modification; see Appendix C4 for a discussion.

We have also added more detail to the text formerly found in Appendix C3, and moved it to it's own new Appendix C4:

> **C4    Avoiding a low injection height by revising the LW mass extinction coefficient tuning**
>
> As alluded to in Section 3.5 and Appendix C3, there is some degeneracy between $b_{LW,ash}$ and $b_{LW,SO2}$ for controlling the initial heating of the aerosol plume, as well as degeneracy between $b_{LW,sulfate}$ and $b_{LW,SO2}$ for controlling the stratospheric temperature anomalies during the first few months post-injection. This makes the manual process of iteratively tuning the parameters more laborious. In the present case, it also results in the implementation of a unusually low initial injection height of $\mu = 14$ km. Specifically, we did not tune $b_{LW,SO2}$ along with $b_{LW,ash}$ and instead needed to compensate for the aggressive early plume lofting by lowering $\mu$.
>
> The tuning process would be easier, and a higher initial injection height of 18-20 km could be more easily supported, if the degeneracy between these three extinction parameters were removed. We  suggest having the SO₂ tracer instead behave as a radiatively passive tracer, acting only as the vehicle for sulfate production, by setting $b_{LW,SO2} = 0$ and $b_{SW,SO2} = 0$. In this case, the LW mass extinction coefficients for ash and sulfate would  be independent knobs for the lofting height, and long-term temperature anomalies, respectively. We would consider this tuning choice an improvement of the parameterization.

We have also added a more explicit pointer to this discussion in Section 2.1:

> After tuning the model with these considerations in mind, we use the even lower value of $\mu = 14$ km, which we found to result in a realistic settling altitude for the sulfate tracer distribution. . The need for this exceptionally low injection height is due to an overly aggressive heating of the

initial plume given our parameter choices, which is discussed further in Section 3.5 and Appendix C4.

**1.5. Comment 4**

**RC:** *In a similar vein, the 30-day e-folding time used for the Pinatubo SO2 is considered fairly uncertain—faster e-folding times (23±5 days or 25±5 days depending on choice of dataset) have been proposed (Guo et al., 2004, https://doi.org/10.1029/2003GC000654).*

AR: We thank the reviewer for catching this error. We do indeed use a 25-day e-folding time for SO2, as informed by Guo et al. 2004. This figure was presented correctly in Table 1, but was later quoted incorrectly as 30 days in the text in both Section 3.1 and Section 3.2. We have corrected these mistakes in the text to instead read "25".

---

## Author Comment (AC3)

**Authors' Response to the Editor of**

**HSW-V v1.0: localized injections of interactive volcanic aerosols and their climate impacts in a simple general circulation model**

Joseph Hollowed, Christiane Jablonowski, Hunter Y. Brown, Benjamin R. Hillman, Diana L. Bull, and Joseph L. Hart
*egusphere-2024-335*
* * *
EC: *Editor Comment*,     AR: Author Response,     ☐ Manuscript Text

**1. Editor Comments**

**1.1. Comment 1**

**EC:** *In particular, please note that for your paper, the following requirement has not been met in the Discussions paper:*

- *The main paper must give the model name and version number (or other unique identifier) in the title.*

*Please add the name and version number of the model used (E3SMv2) to the title of your manuscript*

AR: Rather than include the name of the climate model employed in our experiments (E3SMv2), we decided to give the specific model configuration and parameterization set presented in this work a name, "HSW-V". This model name is now present in the manuscript title with the version number "v1.0". The full article title is now *HSW-V v1.0: localized injections of interactive volcanic aerosols and their climate impacts in a simple general circulation model*.

**EC:** *Your reference list includes works "in preparation". Such works can be cited upon submission if being available to the reviewers. They should not be cited in the final, accepted manuscript, unless published, accepted for publication, or available as preprint with a DOI.*

AR: We have removed references to works in preparation.

**EC:** *Regarding figure 8: Please ensure that the colour schemes used in your maps and charts allow readers with colour vision deficiencies to correctly interpret your findings. Please check your figures using the Coblis – Color Blindness Simulator (https://www.color-blindness.com/coblis-color-blindness-simulator/) and revise the colour schemes accordingly with the next file upload.*

AR: We thank the editor for the careful consideration of this figure's visibility. We have uploaded Figure 8 to the Coblis Color Blindness Simulator as suggested. We find that the color of the cooling rate contours in panel (a) are difficult to distinguish for both Red-Blind/Protanopia and Monochromacy/Achromatopsia. However, we do not think that these contours would be confused with those of AOD even in the monochromatic case, since the AOD contours in this panel are white, and the cooling rate contours are clearly specified in the figure legend. We have updated the figure caption to make this more explicit to avoid confusion:

**Figure 8.** **(a.)** Zonal-mean AOD in the latitude-time plane for the first 90 days post-injection. Overplotted is the cooling rate imposed on the lowest model level by shortwave extinction every 0.15 K day1 in solid red contours. **(b)** Logarithmic zonal-mean AOD over 1000 days. The 0.1 and 0.001 AOD lines are in bold. Cooling rates are not overplotted in this panel. A faint dotted line shows the equator, and a black triangle shows the time and latitude of the injection.